# Tankyrase inhibition preserves osteoarthritic cartilage by coordinating cartilage matrix anabolism via effects on SOX9 PARylation

Sukyeong Kim [1,2,6], Sangbin Han [1,2,6], Yeongjae Kim[1,2], Hyeon-Seop Kim[1,2], Young-Ran Gu[1,2], Donghyun Kang [1,2], Yongsik Cho[1,2], Hyeonkyeong Kim[1,2], Jeeyeon Lee[1,2], Yeyoung Seo[1,2], Moon Jong Chang[3], Chong Bum Chang[4], Seung-Baik Kang [3] & Jin-Hong Kim [1,2,5]*

Osteoarthritis (OA) is a prevalent degenerative disease, which involves progressive and irreversible destruction of cartilage matrix. Despite efforts to reconstruct cartilage matrix in osteoarthritic joints, it has been a difficult task as adult cartilage exhibits marginal repair capacity. Here we report the identification of tankyrase as a regulator of the cartilage anabolism axis based on systems-level factor analysis of mouse reference populations. Tankyrase inhibition drives the expression of a cartilage-signature matrisome and elicits a transcriptomic pattern that is inversely correlated with OA progression. Furthermore, tankyrase inhibitors ameliorate surgically induced OA in mice, and stem cell transplantation coupled with tankyrase knockdown results in superior regeneration of cartilage lesions. Mechanistically, the pro-regenerative features of tankyrase inhibition are mainly triggered by uncoupling SOX9 from a poly(ADP-ribosyl)ation (PARylation)-dependent protein degradation pathway. Our findings provide insights into the development of future OA therapies aimed at reconstruction of articular cartilage.

[1] Center for RNA Research, Institute for Basic Science, 08826 Seoul, South Korea. [2] Department of Biological Sciences, College of Natural Sciences, Seoul National University, 08826 Seoul, South Korea. [3] Department of Orthopaedic Surgery, Seoul National University College of Medicine, Boramae Hospital, 07061 Seoul, South Korea. [4] Department of Orthopaedic Surgery, Seoul National University Bundang Hospital, 13620 Seongnam, South Korea. [5] Interdisciplinary Program in Bioinformatics, Seoul National University, 08826 Seoul, South Korea. [6] These authors contributed equally: Sukyeong Kim, Sangbin Han. *email: jinhkim@snu.ac.kr

Osteoarthritis (OA) is the most common of all arthritis and one of the leading causes of chronic disability in elderly populations[1]. The prevalence of OA is increasing worldwide, imposing a tremendous medical and socioeconomic burden. OA is primarily characterized with the loss of proteoglycan content and degradation of collagen in the cartilage matrix[2]. The degeneration of cartilage matrix during OA development eventually causes failure in the load-bearing functions of articular cartilage.

Disease-modifying OA therapy essentially aims at prolonged and functional repair of articular cartilage[3,4]. Cartilage repair is steered by chondrogenic differentiation of resident progenitor cells and extracellular matrix (ECM) anabolism by the differentiated chondrocytes[5]; however, these abilities generally decline with aging and disease progression, leaving a marginal repair capacity in osteoarthritic joints[2]. To compensate for the lack of intrinsic repairing abilities, stem cell-based therapies relying on the enrichment of mesenchymal stem cell (MSC) population via marrow stimulation or direct transplantation into the damaged region have been widely attempted[6–8]. These cells, however, tend to rapidly lose chondrogenic features after their differentiation, accompanied with the cessation of cartilage matrix molecule synthesis[9].

The extensive molecular exploration of chondrogenesis has led to the identification of several key factors involved in the regulation of chondrocytic phenotypes[10–12]. In particular, SOX9 was identified as a master transcription factor of chondrogenesis expressed in the developing cartilage[13–15]. SOX9 is most actively expressed during chondrogenic differentiation of multipotent mesenchymal progenitors and its activity is repressed during the hypertrophic maturation of chondrocytes in growth plates[16,17]. SOX9 remains expressed in chondrocytes of articular cartilage and regulates ECM homeostasis through the regulation of the expression of major constituents of the cartilage-specific matrix, including type II, type IX, and type XI collagen as well as aggrecan[18]. However, SOX9 transcriptional activity is downregulated in osteoarthritic chondrocytes, thereby resulting in the decrease in the matrix anabolism of articular cartilage[19]. In fact, forced overexpression of SOX9 in MSCs prior to its transplantation in cartilage defects has demonstrated its beneficial roles in promoting chondrogenic differentiation[20,21]. Despite the disease-modifying potentials of SOX9, no effective ways have been described to fully activate endogenous SOX9 for the purpose of cartilage repair.

Tankyrase (encoded by TNKS or TNKS2) belongs to the poly (ADP-ribose) polymerase (PARP) superfamily[22,23]. Tankyrase interacts with target proteins through their unique ankyrin repeat protein-interaction domain and catalyzes the addition of ADP-ribose moieties onto target proteins by using $NAD^+$ as the substrate[24]. This post-translational modification process, termed as poly(ADP-ribosyl)ation (PARylation), is known to regulate the activity and stability of substrate proteins[22]. At present, several highly specific and potent tankyrase inhibitors are available that may provide opportunities to develop therapies to modulate tankyrase activity[24].

Here, we demonstrate that tankyrase inhibition substantially stimulates pro-anabolic pathways in articular cartilage. Tankyrase inhibition augments cartilage-specific ECM synthesis by chondrocytes and enhances the chondrogenic differentiation of MSCs. The controlled delivery of tankyrase inhibitors to knee joints ameliorates surgically induced OA in mice. MSCs with tankyrase knockdown exhibit superior regeneration capacity with robust ECM accumulation upon their transplantation to cartilage defects. The pro-anabolic effect of tankyrase suppression in cartilage is mainly mediated by the activating cartilage-signature transcriptional network via decoupling of SOX9 from PARylation-dependent degradation.

## Results

**Tankyrase is a regulator of cartilage matrix anabolism.** To screen for a key regulatory factor that could be targeted to stimulate cartilage matrix anabolism, we conducted genetic analysis on transcriptomes of mouse reference populations by using post hoc factor analysis. First, we assessed the transcriptional variance in the cartilage tissues of 16 strains of BXD mice[25]. We noted that among 21 cartilage matrix genes listed up by Heinegard and Saxne[26], 14 cartilage matrix genes showed strong positive correlation in their transcript abundance (Fig. 1a). These high correlations were absent in organs without cartilaginous functions, such as bone femur, kidney, lung, and brain (Supplementary Fig. 1a). A genome-wide co-expression analysis by using BXD cartilage transcriptome datasets similarly resulted in the identification of a highly correlated cluster enriched with cartilage matrix genes (Supplementary Fig. 1b). We then attempted to extract a common axis underlying cartilage anabolism by performing a principal component analysis on 14 highly intercorrelated cartilage matrix genes (see black box in Fig. 1a). The first axis identified (Factor 1) essentially reflects the state of cartilage matrix anabolism (Fig. 1b). Next, we used the Gene Ontology (GO) Resource to annotate the 16,074 genes included in the cartilage transcriptome, of which 6824 belonged to the GO terms: "catalytic activity", "DNA-binding transcription factor activity", or "regulation of signaling", which can potentially have regulatory functions on the collective expression of a set of genes. We then computed Pearson's correlation coefficients between Factor 1 and these 6824 genes with potential regulatory functions. Tankyrase showed striking negative correlations with the anabolic axis and with individual cartilage matrix genes and was therefore investigated further (Fig. 1b, c and Supplementary Fig. 1c, d).

We examined the potential regulatory role of tankyrase in cartilage anabolism. Knockdown of both Tnks and Tnks2 collectively induced the expression of cartilage-specific matrix genes in primary cultured mouse chondrocytes (Fig. 1d–f). On the other hand, the individual knockdown of Tnks or Tnks2 failed to increase the cartilage matrix anabolism, suggestive of the redundant roles of TNKS and TNKS2 in this regulation (Fig. 1e). Treatment with XAV939 or IWR-1, highly specific and potent TNKS/2 inhibitors[27], also increased the expression of cartilage-specific matrix genes in chondrocytes (Fig. 1g–j). However, the PARP1/2 inhibitor, ABT-888, failed to increase their expressions (Fig. 1i, j).

To comprehensively elucidate the effect of tankyrase inhibition at the whole transcriptome level, we performed RNA sequencing for chondrocytes treated with siRNAs targeting Tnks and Tnks2, XAV939, or IWR-1 (Supplementary Fig. 2a, b). Of the 711 differentially expressed genes (DEGs) identified in this analysis, the 527 genes (74.12%) were consistently upregulated or downregulated in all three tankyrase inhibition groups compared with their respective control groups (Supplementary Fig. 2c). GO analysis of the commonly upregulated genes in response to tankyrase inhibition revealed a strong association with terms related to cartilage development (Supplementary Fig. 2d). Next, we generated a comprehensive list of cartilage-signature genes by utilizing public transcriptome datasets[18]. Tankyrase inhibition induced strong transcription of key cartilage-identity genes (Supplementary Fig. 2e). In addition, gene set enrichment analysis (GSEA) revealed that cartilage-signature genes were positively enriched in the whole transcriptome obtained from chondrocytes treated with siTnks and siTnks2 or tankyrase inhibitors, XAV939 and IWR-1 (Fig. 1k, l). Thus, tankyrase inhibition promotes cartilage matrix anabolism and strengthens the overall chondrogenic features in chondrocytes.

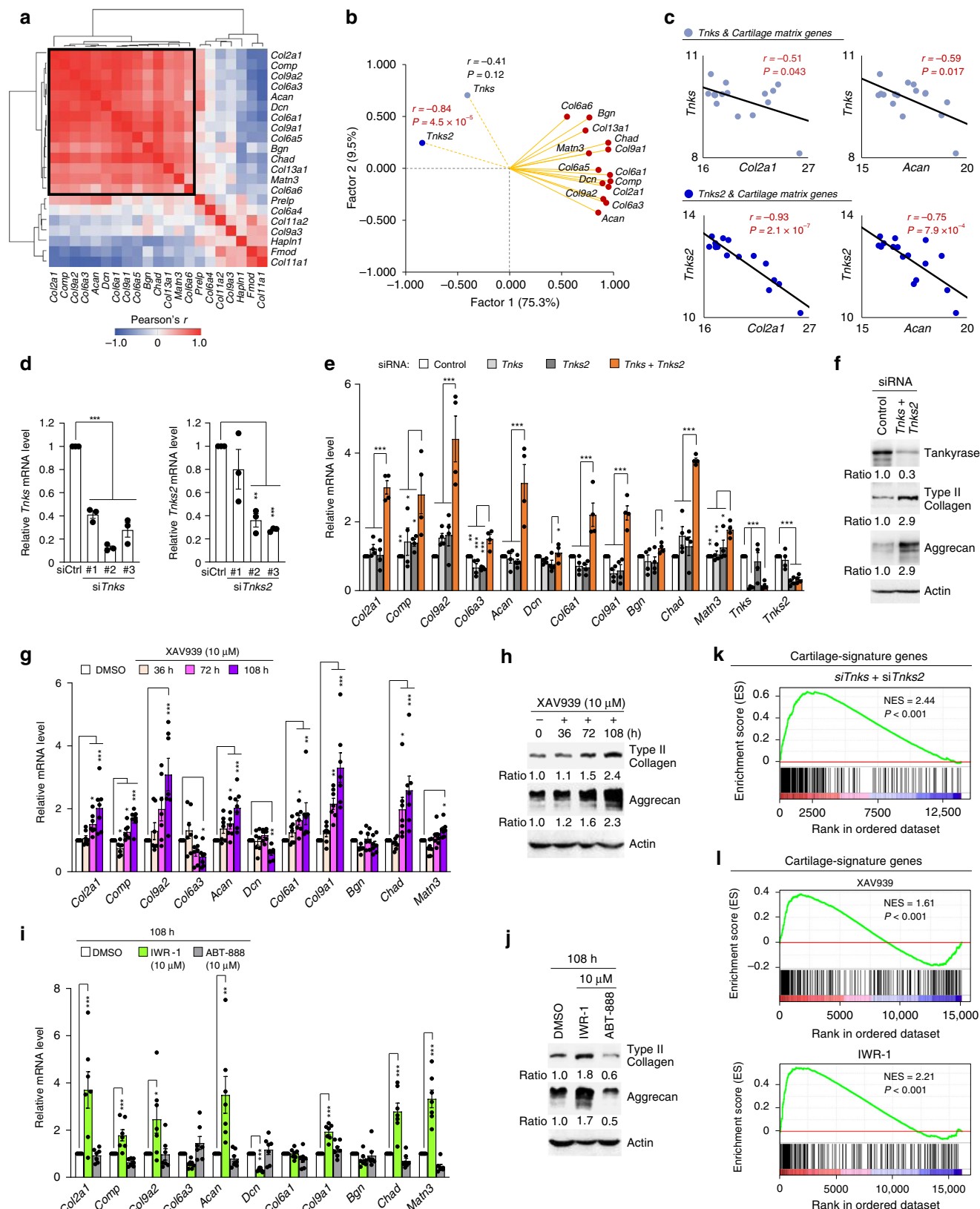

**SOX9 interacts with tankyrase**. To understand the molecular mechanism underlying the effect of tankyrase inhibition on cartilage anabolism, we aimed to identify tankyrase substrates responsible for the regulation of cartilage matrix genes. Axin, a well-established target of tankyrase, is subjected to proteasomal degradation upon PARylation-dependent ubiquitination[27]. In fact, $Tnks2$ was the 12th highest among the 431 genes enriched for the GO term "Wnt signaling pathway" in terms of Pearson's correlation coefficients with the cartilage anabolic axis (Supplementary Table 1). Tankyrase inhibition caused a minor reduction in the β-catenin protein level in chondrocytes (Fig. 2a); the effects of tankyrase inhibition on β-catenin stability and activity were

**Fig. 1** Identification of tankyrase as a regulator of cartilage anabolic axis. **a** Heatmap of Pearson's correlation coefficients of transcript levels for cartilage matrix genes from 16 BXD mouse strains. **b** Factor-loading plot of 14 cartilage matrix genes in cartilage of the 16 BXD mouse strains in terms of transcript abundance, with *Tnks* and *Tnks2* added to the plot. Pearson's *r* and *P* value displayed next to *Tnks* or *Tnks2* point represent correlation strength between Factor 1 and *Tnks* or *Tnks2*. **c** Correlation between *Tnks* or *Tnks2* and *Col2a1* or *Acan* mRNA levels in the 16 BXD mouse strains. **d** Knockdown efficiency of various *Tnks* and *Tnks2* siRNAs in primary cultured mouse chondrocytes (*n* = 3). si*Tnks* #2 and si*Tnks2* #3 were used throughout this study. **e–j** mRNA and protein levels of cartilage-specific matrix genes in mouse chondrocytes treated with (**e, f**) control or *Tnks* and *Tnks2* siRNAs (**e**; *n* = 4) or (**g–j**) drugs (**g, i**; *n* = 7). *Col6a5*, *Col6a6*, and *Col13a1* mRNAs were undetected. **k** GSEA of cartilage-signature genes in mouse chondrocytes transfected with si*Tnks* and si*Tnks2* compared with control siRNA. **l** GSEA of cartilage-signature genes in chondrocytes treated with tankyrase inhibitors compared with vehicle. Cartilage-signature genes are listed in Supplementary Table 9. Genes upregulated in mouse chondrocytes compared with mouse embryonic fibroblasts were selected as cartilage-signature genes. **d, e, g, i** Data represent means ± s.e.m. *P < 0.05, **P < 0.01, ***P < 0.001 by ANOVA. **f, h, j** The relative protein band intensity is indicated. Full-size immunoblot images are provided in Supplementary Fig. 9

more pronounced in chondrocytes treated with exogenous Wnt ligands (Fig. 2b–e). Therefore, we sought to determine whether the effect of tankyrase inhibition on cartilage anabolism was associated with its effect on Wnt/β-catenin signaling inhibition. siRNA-mediated knockdown of β-catenin had no significant effect on the expression of cartilage matrisome (Fig. 2f). Similarly, treatment with Dkk-1, which antagonizes canonical Wnt ligands[28], or IWP-2, a porcupine inhibitor that blocks the secretion of canonical and noncanonical Wnts[29], did not affect the expression of *Col2a1* and *Acan* (Fig. 2g, h). Moreover, the upregulation of *Col2a1* and *Acan* induced by XAV939 treatment was not affected by siRNA-mediated knockdown of β-catenin in chondrocytes (Fig. 2i). Therefore, in the absence of exogenous Wnt ligands, the effect of tankyrase inhibition on cartilage anabolism was independent of Wnt/β-catenin signaling.

We then examined the role of β-catenin in regulating cartilage anabolism in the presence of exogenous Wnt ligands. Treatment of chondrocytes with Wnt-3a, a canonical Wnt ligand, significantly reduced the expression of *Col2a1* and *Acan* (Fig. 2g, h). The addition of recombinant Dkk-1 effectively abolished the Wnt-3a-mediated suppression of these chondrogenic genes. Meanwhile, IWP-2 did not rescue the expression of *Col2a1* and *Acan*, which is consistent with its inability to block exogenously added Wnt ligands. Therefore, in the presence of abundant Wnt ligands, tankyrase inhibition may exert an additional effect to promote cartilage matrix anabolism in a Wnt/β-catenin signaling-dependent manner.

Nonetheless, because our study was mainly conducted in a context where β-catenin signaling plays a marginal role in cartilage matrix gene regulation, we hypothesized that there may be an alternative mechanism, whereby tankyrase plays a regulatory role in cartilage-specific anabolic signaling. To find tankyrase-binding substrates that regulate cartilage matrix anabolism, we performed liquid chromatography–tandem mass spectrometry (LC–MS/MS) analysis for the proteome co-immunoprecipitated with the endogenous tankyrase in chondrocytes (Fig. 3a and Supplementary Data 1). We considered proteins that are detected in more than one biological replicate as putative tankyrase-interacting proteins (Fig. 3b). Among these binding partners, candidate substrates were further screened by using a tankyrase-targeting score (TTS) system (Supplementary Data 2)[30]. Ingenuity-pathway analysis (IPA) revealed four candidate proteins above the TTS cutoff (≥0.385) that fell into the *chondrogenesis* category (Fig. 3c). IUPred disorder score[31] was used to filter unlikely targets, wherein tankyrase-binding motifs are positioned in a highly structured region (Fig. 3d). SOX9, which is known as the master transcription factor of chondrogenesis[32], was highly predicted as a substrate candidate, exhibiting both high TTS and disorder scores. Endogenous interactions between tankyrase and SOX9 in chondrocytes were confirmed by co-immunoprecipitation assay and in situ proximity ligation assay (PLA) (Fig. 3e, f). Moreover, our cell-based

assay indicated that SOX9 binds to both TNKS and TNKS2 (Fig. 3g). The two tankyrase-binding domains (TBDs) of SOX9, designated as TBD1 and TBD2, are highly conserved among vertebrates (Fig. 3h). Based on structural simulations, TBD1 and TBD2 peptides fit into the binding pocket located central to the ankyrin repeat cluster (ARC) IV domain of tankyrase where known substrates, SH3 domain-binding protein (3BP2) and myeloid cell leukemia sequence 1 protein (MCL1), are aligned (Fig. 3i). The deletion of either TBD1 or TBD2 resulted in the reduction in the binding affinity of SOX9 for tankyrase (Fig. 3j), while simultaneous deletion of both TBDs nearly abolished this association (Fig. 3k).

**Tankyrase PARylates SOX9 and promotes its degradation**. We investigated whether tankyrase binding to SOX9 is coupled to PARylation of SOX9. Wild-type SOX9 underwent extensive PARylation, whereas SOX9 mutant missing both TBDs exhibited a markedly reduced PARylation level (Fig. 3l). Consistently, the extent of SOX9 PARylation was also reduced by the pharmacological inhibition of tankyrase activity through treatment with XAV939 (Fig. 3m). Tankyrase-dependent PARylation is generally linked to the degradation of substrate proteins[22]. In fact, tankyrase inhibition promoted SOX9 protein expression in chondrocytes (Fig. 3n, o). The SOX9 TBD mutant showed augmented stability compared with wild-type SOX9 (Fig. 3p, q). Taken together, the disruption of the physical interactions between tankyrase and SOX9 and the consequent abolishment of SOX9 PARylation results in the stabilization of SOX9.

We then investigated the mechanism whereby tankyrase-mediated PARylation of SOX9 affects its polyubiquitination and subsequent proteasomal degradation. HEK293 cells expressing FLAG-tagged human SOX9 were treated with XAV939 or vehicle. Overexpressed SOX9 underwent extensive polyubiquitination, whereas XAV939 treatment nearly abolished the ubiquitination of SOX9 in HEK293 cells (Fig. 3r), indicating that tankyrase-mediated PARylation is an essential modification that leads to SOX9 ubiquitination. To provide further evidence for a functional interaction between SOX9 and tankyrase in a physiological context, we examined the ubiquitination of endogenous SOX9 in primary chondrocytes, with or without tankyrase inhibition. Using a tandem ubiquitin-binding entity (TUBE) directed against polyubiquitin moieties, endogenous proteins with polyubiquitin chains were immunoprecipitated. Western blotting analysis revealed that endogenous SOX9 undergoes polyubiquitination and is degraded by the proteasome in chondrocytes, as MG132 treatment led to the accumulation of SOX9 and its polyubiquitinated forms (Fig. 3s). XAV939 treatment effectively reduced the polyubiquitinated forms of SOX9 to levels comparable with those observed in the absence of MG132. Together, our results show that SOX9 undergoes degradation through the ubiquitin–proteasome system and this can be effectively reversed by the inhibition of tankyrase activity.

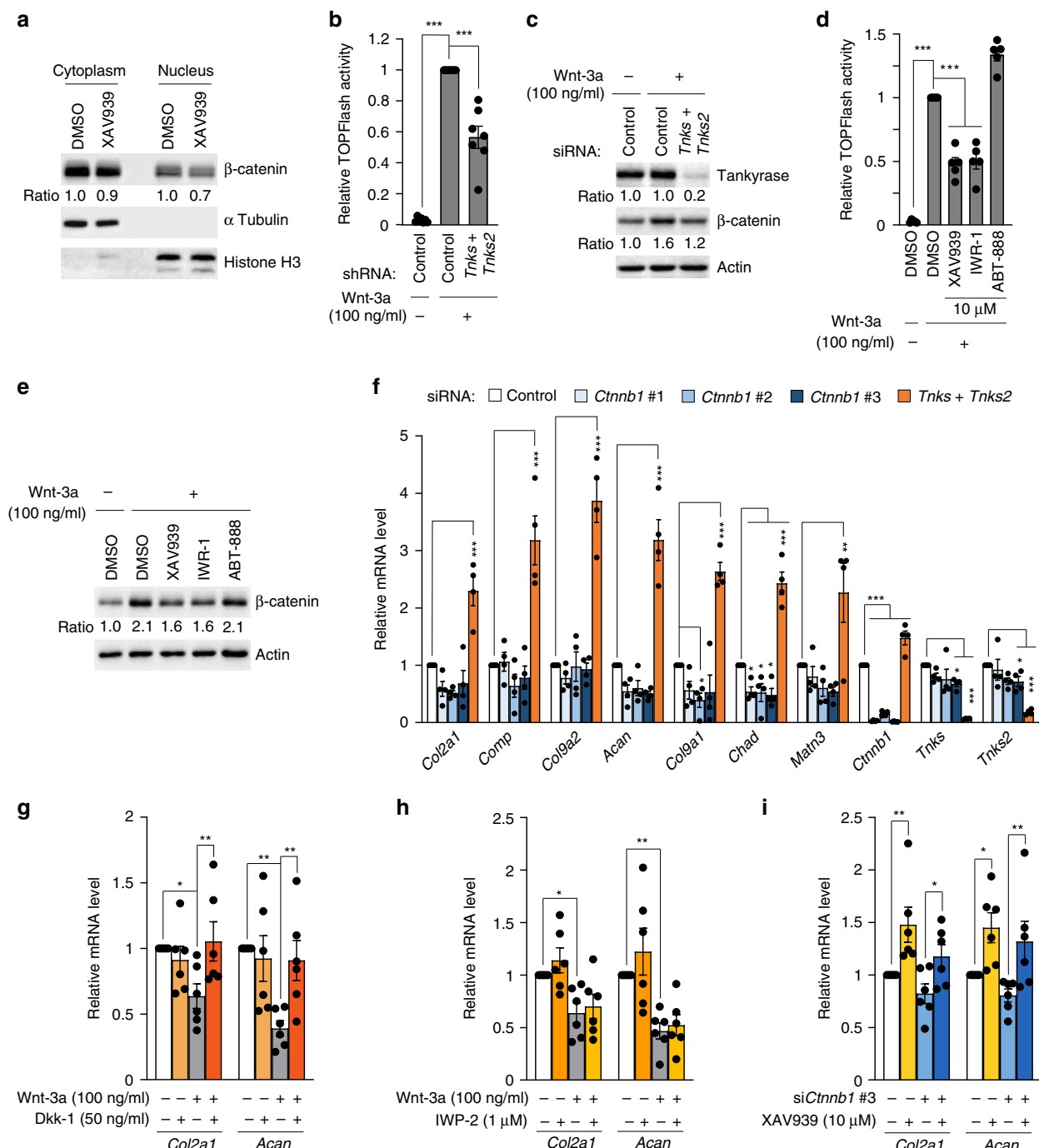

**Fig. 2** Pro-anabolic effect of tankyrase inhibition is mediated by a β-catenin-independent pathway. **a** Cytoplasmic and nuclear fractions from chondrocytes treated with DMSO or XAV939 (10 μM, 72 h) were immunoblotted for β-catenin. The band intensities were normalized with respect to the DMSO-treated control within each fraction. **b** TOPFlash reporter assay in chondrocytes after control shRNA or sh*Tnks* and sh*Tnks2* transfection (*n* = 7). **c** Chondrocyte lysates were immunoblotted for β-catenin after treatment with control or *Tnks* and *Tnks2* siRNAs. **d** TOPFlash reporter assay in chondrocytes following drug treatment (10 μM, 48 h; *n* = 5). **e** Chondrocyte lysates were immunoblotted for β-catenin after treatment with the indicated 10 μM of drugs for 48 h. **b–e** Wnt-3a recombinant protein was added 24 h before harvest. **f** mRNA levels of cartilage matrix genes in chondrocytes transfected with control, *Tnks* and *Tnks2*, or *Ctnnb1* siRNAs (*n* = 4). si*Ctnnb1* #3 was used throughout this study. **g–i** mRNA levels of *Col2a1* and *Acan* in mouse chondrocytes treated with **g** Dkk-1 and Wnt-3a for 72 h (*n* = 6), **h** IWP-2 and Wnt-3a for 72 h (*n* = 6), or **i** control siRNA or si*Ctnnb1* #3 followed by DMSO or XAV939 treatment for 72 h (*n* = 6). **b**, **d**, **f–i** Data represent means ± s.e.m. *$P < 0.05$, **$P < 0.01$, ***$P < 0.001$ by ANOVA. **a**, **c**, **e** The relative protein band intensity is indicated. Full-size immunoblot images are provided in Supplementary Fig. 9

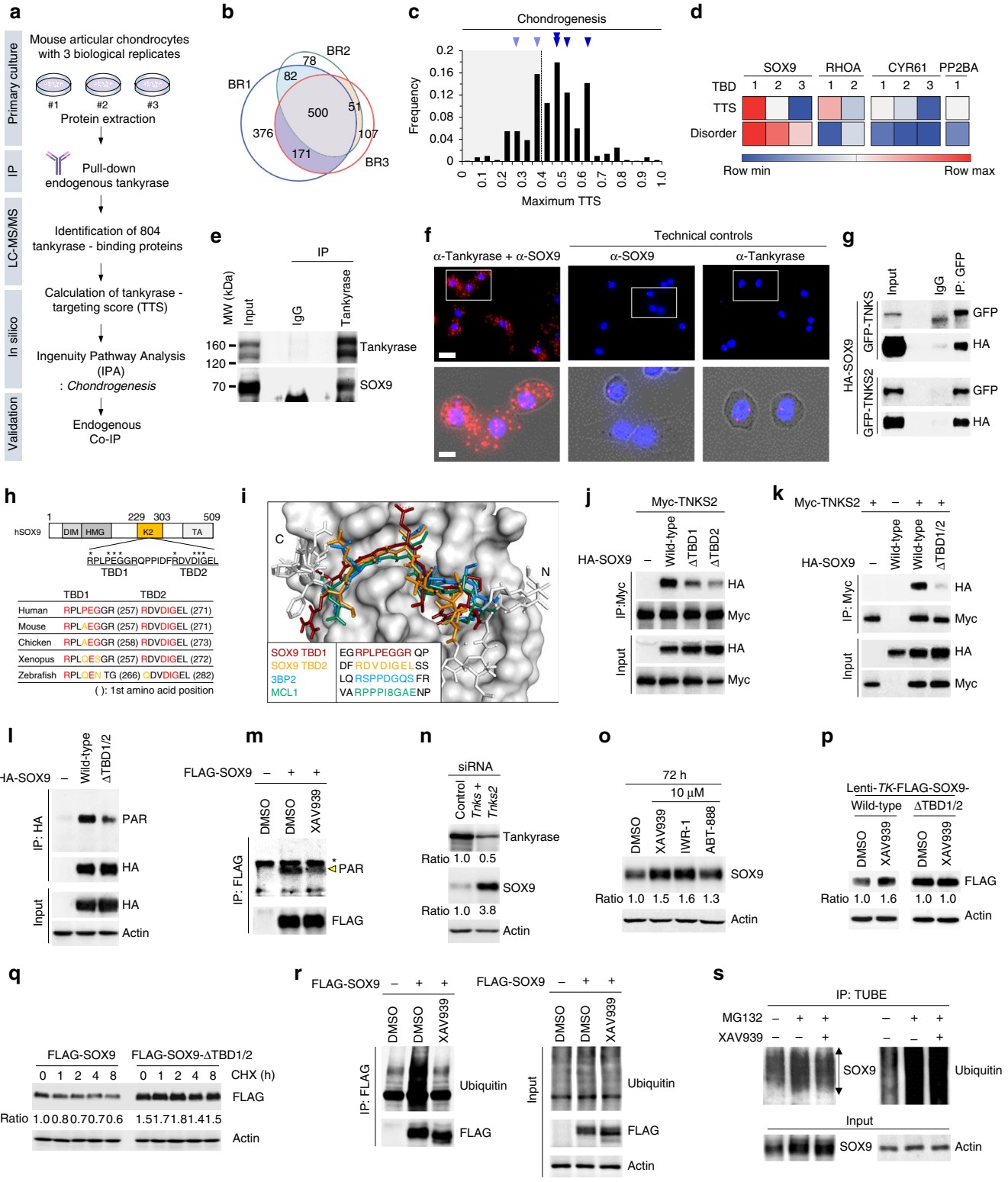

To date, RNF146 is the only known E3 ubiquitin ligase that mediates PARylation-dependent ubiquitination and degradation of substrates[33–35]. In particular, RNF146 is best known to regulate tankyrase-dependent Axin degradation and hence β-catenin stabilization[33]. Consistent with this notion, shRNA or siRNA-mediated knockdown of *Rnf146* effectively reduced TOPFlash activity and β-catenin level (Fig. 4a, b). However, unlike *Tnks* and *Tnks2* double knockdown, *Rnf146* knockdown in chondrocytes failed to increase SOX9 transcriptional

activity, the expression of cartilage matrix genes, or SOX9 protein level (Fig. 4c–f). Our experimental findings were further supported by factor analysis results based on mouse reference populations. A total of 14 intercorrelated cartilage matrix genes exhibited insignificant correlation with *Rnf146* ($r = -0.12$; $P = 0.66$; Fig. 4g). Our data suggest an intriguing possibility that PAR-dependent E3 ligases other than RNF146 may exist and regulate PARylation-dependent SOX9 regulation.

**Fig. 3** Tankyrase interacts with SOX9 and regulates its protein stability. **a** Flowchart of tankyrase substrate identification in chondrocytes. **b** Venn diagram illustrating the overlap of tankyrase-binding proteins identified by three biological replicates (BR) by using LC–MS/MS. **c** Histogram of the maximum TTS of the identified tankyrase-binding proteins. Black down-pointing triangle indicates the bin that includes proteins belonging to the *chondrogenesis* protein set defined by IPA. **d** Heatmap of TTS and disorder score of TBDs from the predicted tankyrase-binding proteins having a maximum TTS of ≥0.385 and belonging to the IPA *chondrogenesis* protein set. The cutoff of 0.385 is the TTS of the tankyrase-binding motifs of mouse AXIN1 and AXIN2. **e** Co-immunoprecipitation of endogenous tankyrase with SOX9 in chondrocytes. MW indicates molecular weights. **f** In situ PLA to detect interaction between endogenous tankyrase and endogenous SOX9 in primary cultured mouse chondrocytes. Red signals indicate the interactions of endogenous tankyrase–SOX9. DAPI was used as a nuclear counterstain. Scale bar: 25 μm (top), 10 μm (bottom). **g** Pull-down assays of GFP-tagged TNKS or TNKS2 with HA-tagged SOX9 in HEK293T cells. **h** Schematic representation of the predicted TBDs in human SOX9 protein (top) and sequence alignment of TBD1 and TBD2 of SOX9 among vertebrates (bottom). Colored letters indicate the consensus amino acid sequence of TBDs. **i** Superimposition of TNKS2:3BP2 and TNKS2:MCL1 complexes with TNKS2 bound to SOX9–TBD1/2. **j** Pull-down assays of Myc-tagged TNKS2 with HA-tagged wild-type SOX9 or TBD1- or TBD2-deleted SOX9 mutants in HEK293T cells. **k** Pull-down assay of TNKS2 with wild-type or TBD1/2-deleted SOX9 in HEK293T cells. **l** PARylation of HA-tagged wild-type or TBD1/2-deleted SOX9 in HEK293T cells. The HA-tagged proteins were pulled down by using anti-HA tag antibodies captured by protein A/G-coupled sepharose beads. In all, 10 μM of MG132 was treated for 6 h before lysis, and 5 μM of ADP–HPD was added to the lysis buffer. **m** PARylation assay of XAV939-treated HEK293 cells. HEK293 cells pretreated with XAV939 (20 μM, 24 h) were washed and incubated with XAV939 (20 μM) and MG132 (20 μM) for 6 h. Arrowhead indicates the position of PARylated SOX9. Asterisk indicates background bands. **n**, **o** SOX9 immunoblots in chondrocytes after **n** si*Tnks* and si*Tnks2* or **o** drug treatment. **p** HEK293 cells stably transduced with FLAG–SOX9–WT or FLAG–SOX9–ΔTBD1/2 were treated with DMSO or XAV939 (10 μM) for 72 h. Their cell lysates were immunoblotted with anti-FLAG antibody. The band intensities were normalized with respect to DMSO-treated control within each stable cell line. **q** Cycloheximide (CHX) chase analysis of FLAG-tagged wild-type or TBD1/2-deleted SOX9 in Myc-TNKS2-overexpressed HEK293 cells. SOX9 was expressed under TK promoter. **r** Ubiquitination assay of FLAG-tagged SOX9 in XAV939-treated HEK293 cells. HEK293 cells pretreated with XAV939 (20 μM, 24 h) were washed and incubated with XAV939 (20 μM) and MG132 (20 μM) for 6 h. **s** Ubiquitination assay of endogenous SOX9 in XAV939-treated mouse articular chondrocytes. Cells pretreated with XAV939 (10 μM, 4 h) were washed and incubated with XAV939 (10 μM) and MG132 (20 μM) for 4 h. Using a TUBE directed against poly-Ubiquitin moieties, endogenous proteome with poly-Ubiquitin chains was immunoprecipitated. **n–q** The relative protein band intensity is indicated. **e**, **g**, **j–s** Full-size immunoblot images are provided in Supplementary Fig. 9

**Tankyrase inhibition enhances SOX9 transcriptional activity.** Next, we used a 4 × 48-p89 SOX9-dependent *Col2a1* enhancer reporter[36] to investigate whether the increase in SOX9 levels with tankyrase inhibition enhances the overall transcriptional activity of SOX9. Double knockdown of *Tnks* and *Tnks2*, and nine different tankyrase-specific inhibitors, specifically increased the transcriptional activity of SOX9 in chondrocytes (Fig. 5a–c). Moreover, the overexpression of wild-type TNKS2 resulted in a marked reduction in the transcriptional activity of SOX9, while the catalytically inactive form of TNKS2 (TNKS2 M1054V) suppressed SOX9 activity to a moderate extent (Fig. 5d). SOX9 target genes[37] were overall upregulated upon tankyrase knockdown or inhibition at the whole transcriptome level (Fig. 5e). Meanwhile, SOX9 is known to bind to its own enhancer and autoregulate its expression[38]. Indeed, *Sox9* transcript levels also increased upon simultaneous knockdown of *Tnks* and *Tnks2* in mouse chondrocytes (Supplementary Fig. 3). Pharmacological inhibition of tankyrase by XAV939 treatment did not have a statistically significant effect on *Sox9* mRNA levels. To further confirm the proposed mechanism that tankyrase regulates SOX9 activity post-transcriptionally, we repeated the luciferase reporter assays by using the SOX9-dependent *Col2a1* enhancer construct in cells constitutively expressing *SOX9* mRNA under the control of a cytomegalovirus (CMV) promoter. Tankyrase inhibition by using siRNAs or drugs increased the transcriptional activity of exogenously expressed SOX9 in HEK293T cells (Fig. 5f, g). Furthermore, point mutations of Arg in the first position to Ala in both TBD1 and TBD2 of SOX9 synergistically enhanced the transcriptional activity of SOX9 (Fig. 5h), suggesting that disruption of the interaction between tankyrase and SOX9 is sufficient to enhance the transcriptional activity of SOX9. Cartilage matrix gene expression induced by tankyrase inhibition was completely abolished by SOX9 knockdown (Fig. 5i, j). Taken together, SOX9 serves as an essential target of tankyrase for the role of tankyrase as an anabolic regulator in chondrocytes.

**Tankyrase inhibition ameliorates OA progression in mice.** Our results suggest that tankyrase may perform a physiological role in

the regulation of cartilage matrix homeostasis. As cartilage homeostasis is disrupted during OA development, we first characterized the expression of tankyrase in articular cartilage of knee OA patients. Tankyrase proteins were markedly upregulated in OA-affected cartilage, but were barely detectable in undamaged regions of osteoarthritic cartilage (Fig. 6a, b). In contrast, type II collagen and aggrecan expression levels were significantly reduced in the damaged regions of human OA cartilage, inversely correlating with the expression pattern of tankyrase (Fig. 6a). We then performed destabilization of the medial meniscus (DMM) surgery in mice as a model of post-traumatic OA. Two weeks after DMM, we observed no distinct damage in the cartilage, but detected robust expression of SOX9 and aggrecan (Supplementary Fig. 4a), consistent with studies that have reported a compensatory synthesis of matrix molecules during the early stage of OA[39,40]. At this stage, tankyrase expression was not detected in the knee joint cartilage of the DMM-operated mice. However, at 8 weeks after DMM surgery, significant damage in the articular cartilage and loss of SOX9 and aggrecan expression were apparent, indicating a mid-to-late stage of OA (Supplementary Fig. 4a, b). A marked increase in tankyrase expression was detected in this condition, consistent with our observations on human OA cartilage.

The increased expression of tankyrase in OA cartilage suggested its possible involvement in OA pathogenesis. Thus, we investigated how tankyrase inhibition affects the expression of OA-associated genes (Supplementary Fig. 5). By utilizing public transcriptome datasets[41], we generated a comprehensive list of OA-associated genes that are upregulated and downregulated in OA patients. Notably, OA-associated genes upregulated in patients were overall repressed in chondrocytes upon tankyrase inhibition (Supplementary Fig. 5a). In contrast, OA-associated genes suppressed in patients were strongly transactivated by tankyrase inhibition (Supplementary Fig. 5b). This inverted pattern of gene expression profiling was evident even at the whole transcriptome level (Fig. 6c, d).

Next, we assessed the in vivo effects of tankyrase inhibition on cartilage matrix homeostasis in surgically induced OA mouse

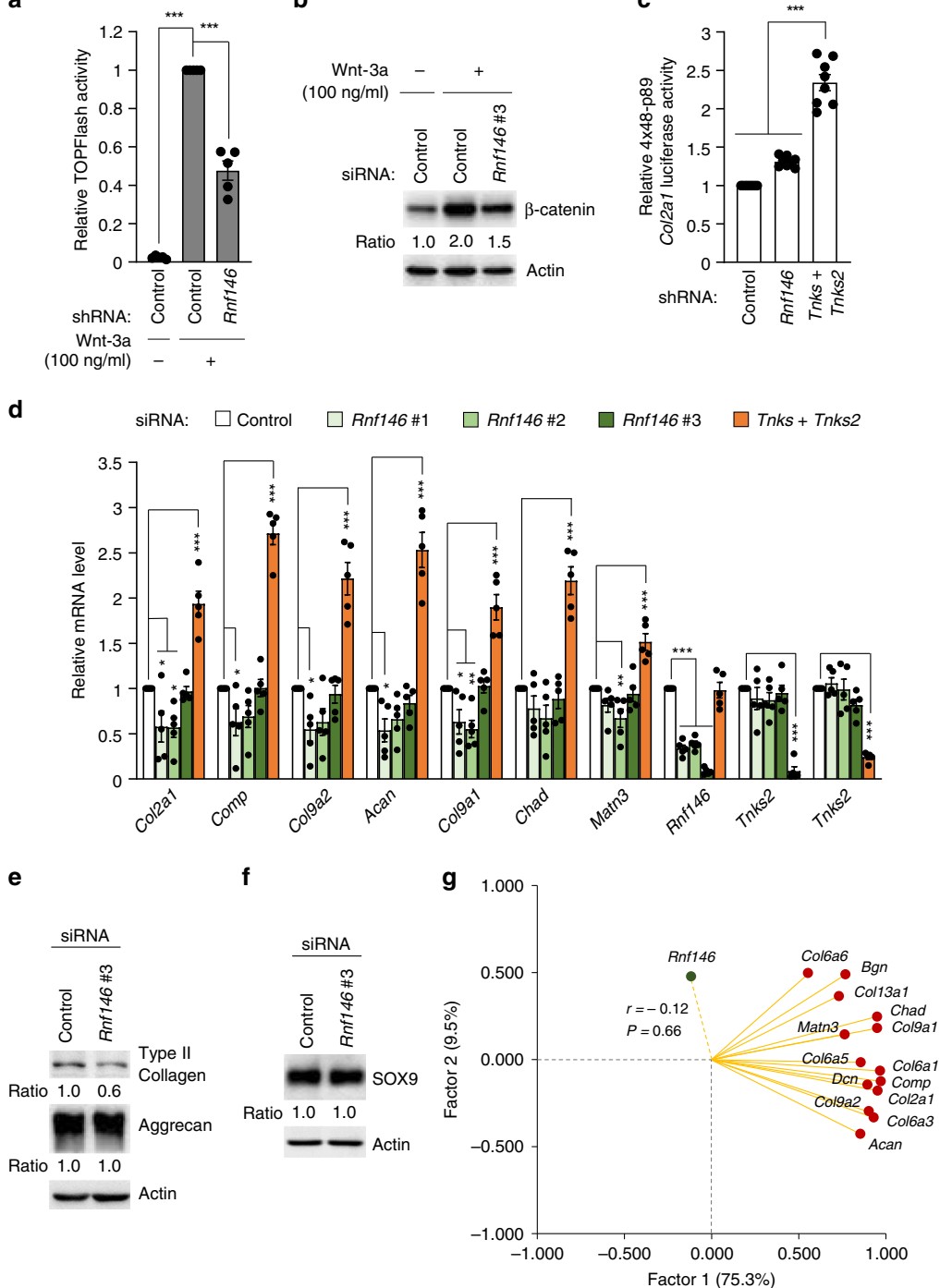

**Fig. 4** RNF146 does not regulate SOX9 activity and cartilage matrix anabolism. **a** TOPFlash reporter assay in chondrocytes after control shRNA or sh*Rnf146* transfection (*n* = 5). **b** Chondrocyte lysates were immunoblotted for β-catenin after treatment with control siRNA or si*Rnf146*. **a**, **b** Recombinant Wnt-3a was added 24 h before harvest. **c** A 4 × 48-p89 SOX9-dependent *Col2a1* luciferase reporter assay in chondrocytes transfected with control shRNA, sh*Rnf146*, or sh*Tnks* and sh*Tnks2* (*n* = 8). **d** mRNA levels of cartilage-specific matrix genes in mouse chondrocytes treated with control siRNA, si*Rnf146*, or si*Tnks* and si*Tnks2* (*n* = 5). **e**, **f** Protein levels of **e** cartilage-specific matrix genes or **f** SOX9 in mouse chondrocytes treated with control siRNA or si*Rnf146*. **g** Factor-loading plot of 14 cartilage matrix genes in the cartilage of 16 BXD mouse strains in terms of transcript abundance, with *Rnf146* added to the plot. **a**, **c**, **d** Data represent means ± s.e.m. $^{*}P < 0.05$, $^{**}P < 0.01$, $^{***}P < 0.001$ by ANOVA. **b**, **e**, **f** The relative protein band intensity is indicated. Full-size immunoblot images are provided in Supplementary Fig. 9

model. For the stable and prolonged delivery of tankyrase inhibitors to mouse knee joints, we used injectable hydrogels made of ascorbyl palmitate[42]. Intra-articular (IA) injection of this hydrogel-based drug delivery system allowed controlled local release of the loaded small molecule to articular cartilage over

9 days (Supplementary Fig. 6a, b). IA administration of hydrogel-mediated XAV939 or IWR-1, the two representative tankyrase inhibitors with different modes of actions, resulted in a significant reduction in the degeneration of cartilage matrix caused by the DMM (Fig. 6e–g). A concomitant increase in type II collagen and

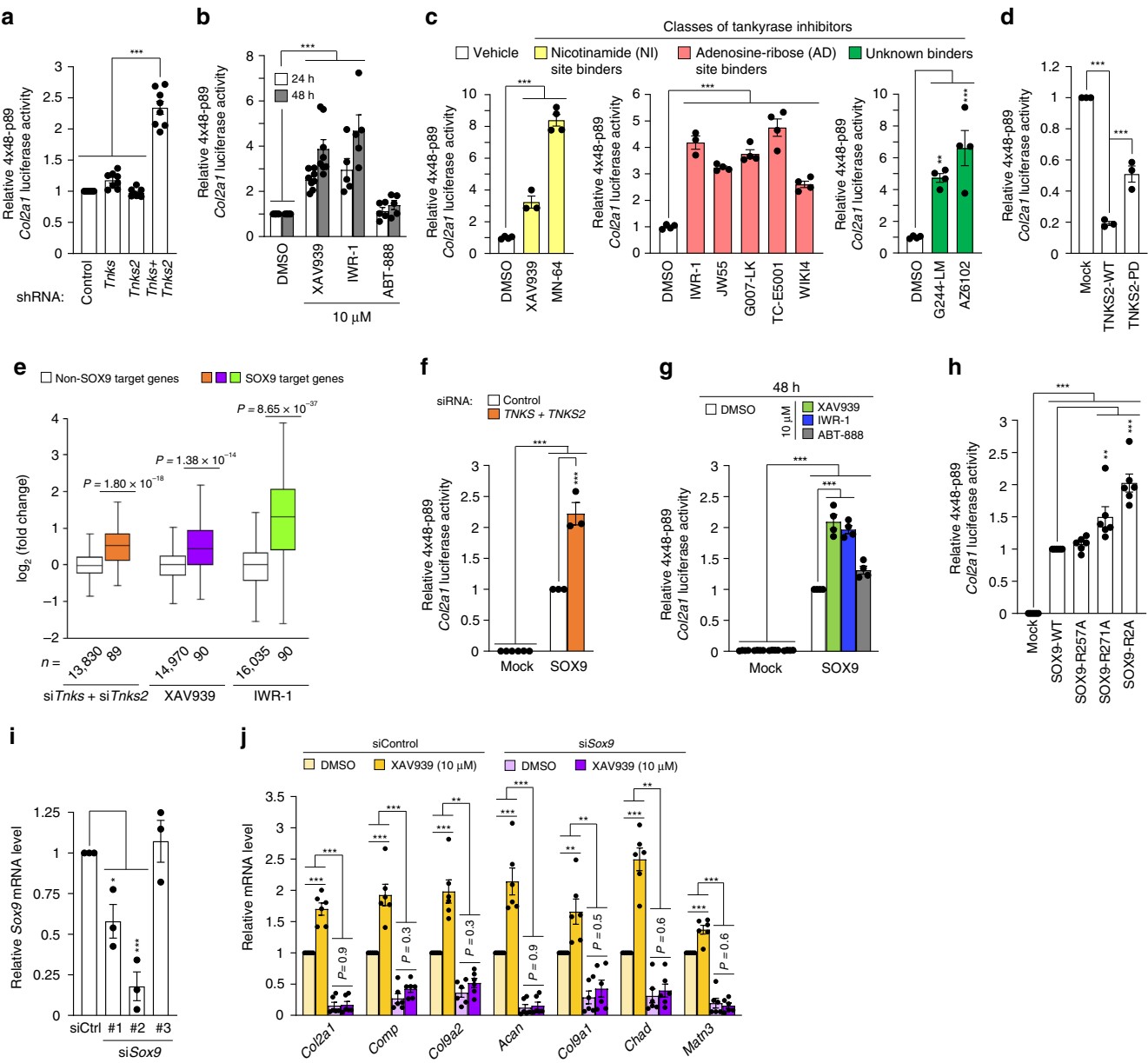

**Fig. 5** Tankyrase inhibition enhances cartilage matrix gene expression in a SOX9-dependent manner. **a**–**c** A 4 × 48-p89 SOX9-dependent *Col2a1* luciferase reporter assay in chondrocytes treated with **a** control shRNA, sh*Tnks*, sh*Tnks2*, or sh*Tnks* and sh*Tnks2* (*n* = 8), **b** XAV939, IWR-1, or PARP1/2 inhibitor ABT-888 (*n* ≥ 5), or **c** 10 μM of various tankyrase inhibitors for 48 h (*n* ≥ 3). **b**, **c** DMSO was used as a negative control. **d** A 4 × 48-p89 SOX9-dependent *Col2a1* luciferase reporter assay in chondrocytes transfected with control mock vector, wild-type *TNKS2* vector, or PARP-dead (PD) *TNKS2* mutant vector (*n* = 3). **e** Box plot of fold changes of SOX9 target genes and other genes (two-tailed *t* test) in chondrocytes. The box represents data from 25 to 75% percentile. The center line in the box represents median, and whiskers extend 1.5 times the interquartile range from the box edges. SOX9 target genes are listed in Supplementary Table 8. **f**, **g** A 4 × 48-p89 SOX9-dependent *Col2a1* luciferase reporter assay in HEK293T cells transfected with mock vector or CMV-driven *SOX9*-expression vector and treated with **f** si*TNKS* and si*TNKS2* (*n* = 3) or **g** DMSO, XAV939, IWR-1, or ABT-888 (*n* = 4). **h** A 4 × 48-p89 SOX9-dependent *Col2a1* luciferase reporter assay in HEK293T cells expressing wild-type SOX9 or SOX9 with tankyrase-binding motif point mutation (*n* = 6). SOX9 R2A mutant has both R257A and R271A mutations. **i** Knockdown efficiency of various *Sox9* siRNAs in primary cultured mouse chondrocytes (*n* = 3). si*Sox9* #2 was used throughout this study. **j** mRNA levels of cartilage-specific matrix genes in mouse chondrocytes transfected with control siRNA or si*Sox9* #2 followed by DMSO or XAV939 treatment for 72 h (*n* = 6). **a**–**d**, **f**–**j** Data represent means ± s.e.m. *P < 0.05, **P < 0.01, ***P < 0.001 by ANOVA

aggrecan was observed (Fig. 6h), and the expression of SOX9 was retained in the cartilage treated with tankyrase inhibitors (Fig. 6i). In addition, we observed that IA delivery of tankyrase inhibitors effectively reduced the production of matrix metalloproteinase 13 (MMP13) and the specific expression of β-catenin in the superficial zone of articular cartilage[43,44] (Supplementary Fig. 6c). These experimental results are in line with the correlation analysis based on mouse reference populations, indicating that tankyrase

exhibits a negative and positive correlation with cartilage matrix genes (Fig. 1b, c) and catabolic regulators (Supplementary Fig. 5c, d), respectively.

Based on the pro-anabolic effect of tankyrase inhibitors, we tested the potential of XAV939 to treat late-stage OA cartilage. In the mouse DMM model, early osteoarthritic lesions were observed 2 weeks after surgery[45], while 70% of mice had reached late-stage OA after 6 weeks from DMM surgery. XAV939

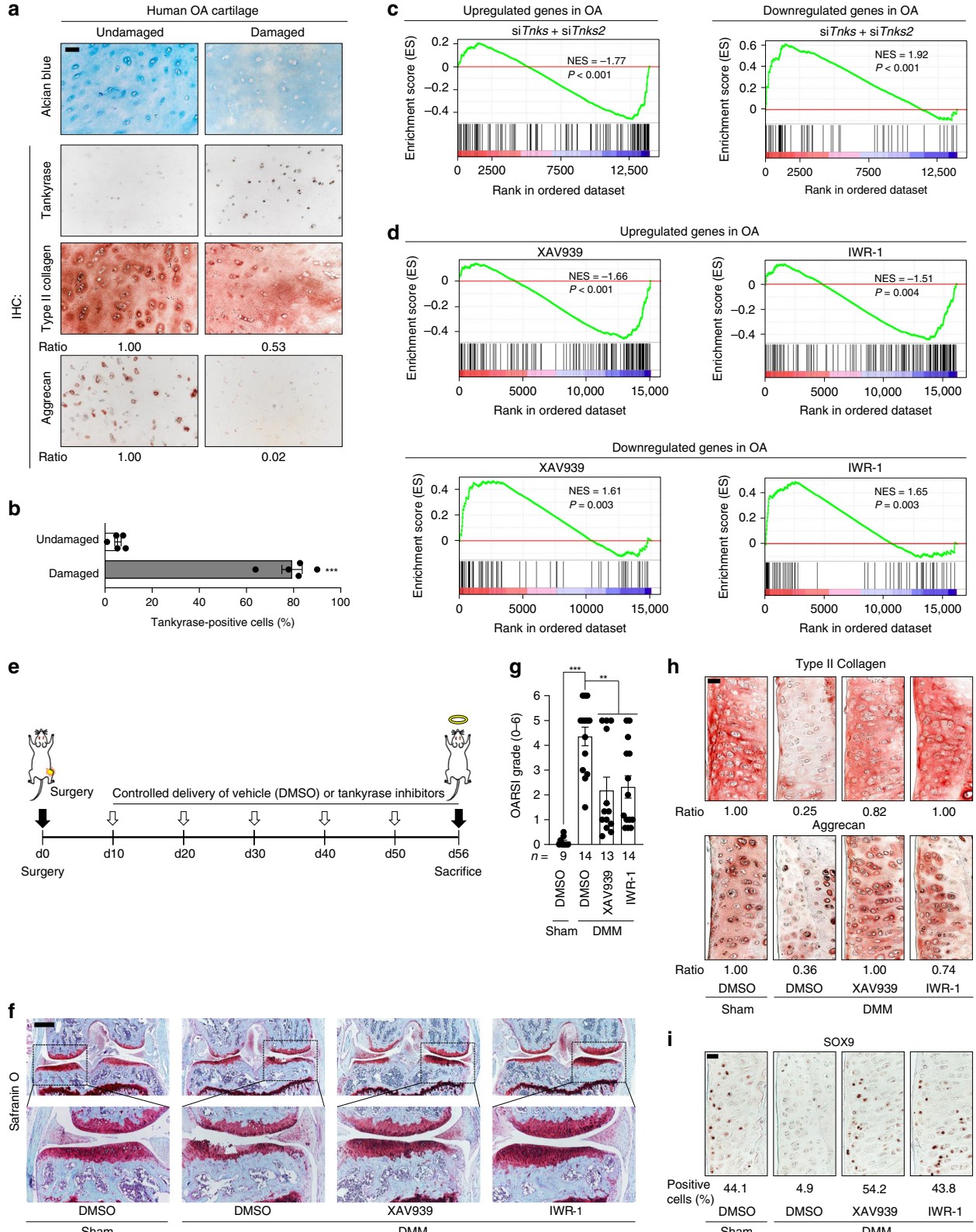

administration for additional 6 weeks resulted in the reduction in the cartilage destruction as compared with the vehicle-treated mice, which experienced further OA progression (Supplementary Fig. 6d–f). Taken together, our results indicate that tankyrase inhibitors effectively ameliorate cartilage destruction in mice through the attenuation of the imbalance between matrix anabolism and catabolism.

**Tankyrase inhibition stimulates chondrogenesis of MSCs**. As mesenchymal progenitor cells are responsible for the regenerative capacity of damaged cartilage[5,46], we investigated the role of tankyrase in the chondrogenic differentiation of MSCs. First, the expression pattern of *Tnks* and *Tnks2* mRNA during chondrogenesis was examined by using a micromass culture of mesenchymal cells isolated from the limb buds of E11.5 mouse

**Fig. 6** Tankyrase inhibition ameliorates OA development in mice. **a**, **b** Tankyrase expression in patients with knee OA. **a** Alcian blue staining and immunostaining of tankyrase and cartilage matrix proteins of undamaged and damaged cartilage regions of knee OA patients. Scale bar: 50 μm. **b** Quantification of tankyrase expression by immunohistochemistry ($n = 5$). **c**, **d** GSEA with OA-signature gene sets in mouse chondrocytes treated with **c** siTnks and siTnks2 versus control siRNA or **d** tankyrase inhibitors versus vehicle. The OA-signature gene sets are listed in Supplementary Tables 10 and 11. Genes upregulated or downregulated in damaged cartilage compared with undamaged cartilage in OA patients were selected as OA-signature genes. **e** Schematic illustration of the DMM model and drug treatment schedule. **f–i** Tankyrase inhibitors protect articular cartilage in surgically induced OA mouse model. Cartilage destruction assessed by **f** Safranin O staining, **g** OARSI grade, and **h** immunostaining of cartilage matrix proteins or **i** SOX9. Scale bar in **f**: 500 μm. Scale bars in **h**, **i**: 25 μm. **g** Data represent means ± s.e.m. ***$P < 0.001$ by Kruskal–Wallis test. **a**, **h**, **i** The relative chromogen intensity or the percentage of immunopositive cells is indicated

embryos. Along with the progression of in vitro chondrogenesis, *Tnks* and *Tnks2* expression were markedly decreased, inversely correlating with the expression of *Acan*, a marker of chondrogenesis (Fig. 7a). We then analyzed a publicly available RNA-Seq dataset[47] generated from pellet cultures of human bone marrow-derived MSCs (hMSCs). *TNKS* and *TNKS2* expression were markedly reduced over the course of the chondrogenic differentiation of hMSCs (Fig. 7b).

Next, we examined how tankyrase inhibition influences the chondrogenic differentiation of MSCs. The tankyrase inhibitors, XAV939 and IWR-1, effectively induced chondrogenic nodule formation in micromass cultures of mouse limb-bud mesenchymal cells (Fig. 7c), enhancing the amount of Alcian blue bound to sulfated glycosaminoglycans (Fig. 7d). The two tankyrase inhibitors significantly promoted *Col2a1* and *Acan* expression, whereas the PARP1/2 inhibitor, ABT-888, failed to increase their expression (Fig. 7e), suggesting that the chondrogenic effect is specific to tankyrase inhibition and not to other PARP family member inhibition. Similarly, both pharmacological inhibition and double knockdown of *TNKS* and *TNKS2* effectively enhanced the chondrogenic differentiation of hMSCs (Fig. 7f–h), with a concomitant upregulation of cartilage-specific proteins, such as type II collagen, aggrecan, and SOX9 (Supplementary Fig. 7a, b).

We next evaluated the effect of tankyrase inhibition on stem cell-based restoration of hyaline cartilage. A full-thickness osteochondral lesion was filled with a fibrin gel containing hMSCs transduced with control or *TNKS* and *TNKS2* shRNAs. After 8 weeks, no positive signals for human-specific mitochondria were detected in osteochondral lesions filled with fibrin gel only, whereas defects transplanted with hMSC-control shRNA or hMSC-*TNKS/2* shRNA displayed strong signals localized in the cytoplasm of transplanted cells (Fig. 7i and Supplementary Fig. 8a). Defects transplanted with hMSC-control shRNA failed to fully recover the organization of hyaline cartilage and exhibited features of fibrocartilage (Fig. 7i, j and Supplementary Fig. 8b, c). However, lesions implanted with hMSC-*TNKS/2* shRNA showed regenerated hyaline cartilage, similar to the articular cartilage. The regenerative tissues derived from hMSC-*TNKS/2* shRNA maintained stable knockdown of tankyrase protein (Fig. 7k) and exhibited robust expression of SOX9 and cartilage-specific matrix proteins (Fig. 7l), compared with the fibrocartilage originated from hMSC-control shRNA. Finally, β-catenin expression was minimal in lesions implanted with hMSC-control shRNA and in those implanted with hMSC-*TNKS/2* shRNA (Fig. 7l).

## Discussion

Recent progress in the understanding of the pathophysiology of OA has led to the identification of promising therapeutic targets, highlighting the potential for the development of disease-modifying osteoarthritis drugs (DMOADs)[48–57]. The identification of MMP13[58] and a disintegrin and metalloproteinase with thrombospondin motifs 5 (ADAMTS5)[55,56] as destructive enzymes for type II collagen and aggrecan led to the active investigation of drugs that may delay OA advancement through

the erosion of cartilage matrix[59,60]. However, agents targeting matrix-degrading processes may be insufficient to restore cartilage ECM and accomplish functional repair of articular cartilage.

We demonstrated here that tankyrase is a key target molecule whose inhibition enhances the expression of cartilage-specific matrisome and contributes to reconstruction of cartilage matrix. Although no functional characterization of tankyrase was previously made in cartilage tissues, there has been an evidence pointing tankyrase as a genetic factor that may potentially affect OA susceptibility. A genome-wide association study indicates that copy-number deletion of tankyrase is associated with a decreased risk for OA[61]. Furthermore, *Tnks* is located within quantitative trait loci associated with spontaneous OA in STR/ORT mice[62]. Our study with quantitative genetic analysis of articular cartilage in BXD mouse reference population illustrates a significant degree of negative correlation between the expression of tankyrase and cartilage-signature genes responsible for ECM construction.

We revealed a previously unrecognized role of tankyrase in the regulation of the stability of SOX9—a master regulator of chondrogenic program[32,63]. In addition to its key role in the regulation of the expression of cartilage matrix genes through its binding to their *cis*-regulatory enhancer elements, SOX9 was shown to be essential for establishing super-enhancer at chondrocyte-identity genes[18]. SOX9 plays an essential role in chondrocyte survival, as SOX9 knockout in chondrocytes triggered extensive apoptosis[64]. SOX9 is downregulated in human osteoarthritic cartilage, and this event is thought to be associated with the cessation of cartilage matrix synthesis and chondrocyte dysfunctions in osteoarthritic cartilage[65,66]. Our findings suggest that tankyrase inhibition may serve as an effective trigger in such OA cartilage to sustain the transcriptional activity of SOX9 and reinforce cartilage-signature gene expression.

Tankyrase associates with multiple target proteins and regulates various signaling pathways, suggestive of the involvement of factors other than SOX9 in the overall effect of tankyrase inhibition on OA pathogenesis. In fact, tankyrase interacts with TRF1[67], Axin[27], NUMA[68], PTEN[69], 3BP2[70], and AMOTs[71], which are implicated in controlling telomere length, Wnt signaling, mitosis, AKT pathway, osteoclastogenesis, and Hippo pathway, respectively. High-throughput proteomic analysis recently revealed extensive tankyrase interactomes, including both enzymatic substrates and nonenzymatic effectors[72,73].

Consistent with the notion that Axin is a well-established tankyrase substrate, tankyrase inhibition effectively reduced β-catenin stability and activity in chondrocytes. Increased levels of β-catenin in OA cartilage have been thought to elicit catabolic profiles in chondrocytes and contribute to cartilage degeneration[74,75]. Various β-catenin inhibitory therapeutics have been used at the preclinical[76–78] and clinical stage[79] (Phase 1: NCT02095548; Phase 2: NCT02536833, NCT03122860; Phase 3: NCT03928184), with promising outcomes in delaying cartilage destruction. The inhibitory role in Wnt/β-catenin signaling may contribute to the chondro-protective effects of tankyrase inhibitors. This notion is further supported by our findings highlighting

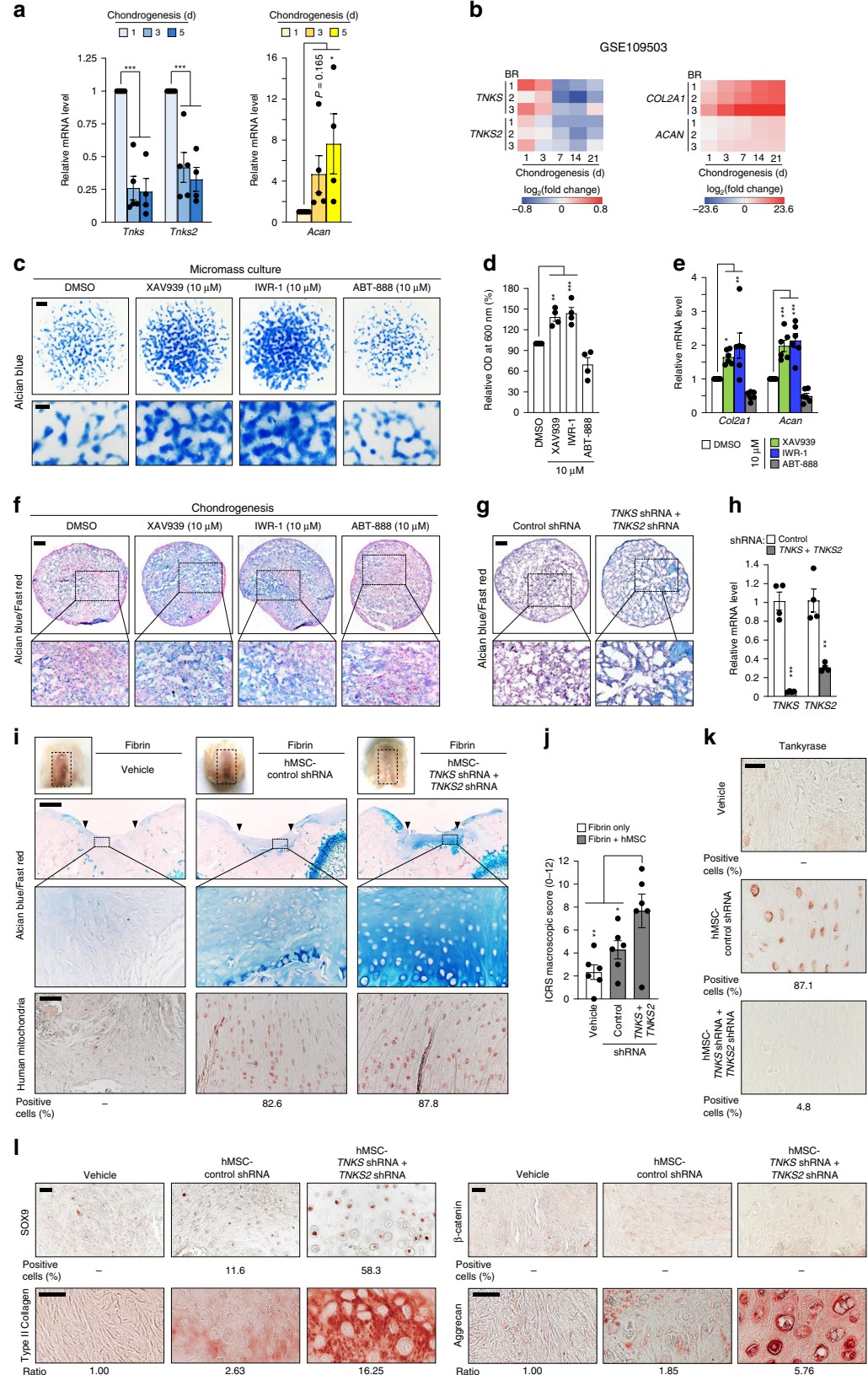

the overall repression of catabolic factors such as MMP13 upon tankyrase inhibition. Therefore, our study suggests that tankyrase inhibition may serve as a potential therapeutic strategy to simultaneously modulate the two key signaling pathways in chondrocytes, SOX9 and β-catenin, in a direction that enhances

cartilage matrix anabolism and inhibits matrix degeneration processes.

Tankyrase-mediated PARylation serves as a signal cue for the ubiquitin conjugation system, which couples to protein degradation[22]. RNF146 is the most prominent E3 ubiquitin ligase

**Fig. 7** Tankyrase inhibition stimulates chondrogenic differentiation of mesenchymal stem cells in vitro and in vivo. **a** *Tnks*, *Tnks2*, and *Acan* mRNA levels in micromass-cultured limb-bud mesenchymal cells ($n \geq 4$). **b** Fold-change heatmaps of *TNKS*, *TNKS2*, *COL2A1*, and *ACAN* in pellet cultures of hMSC relative to day 0. BR indicates biological replicates. **c** Alcian blue staining and **d** absorbance quantification of micromass-cultured limb-bud mesenchymal cells treated with the indicated drugs for 3 days ($n = 4$). Scale bar: 1 mm (top), 300 μm (bottom). **e** *Col2a1* and *Acan* mRNA levels in micromass-cultured limb-bud mesenchymal cells ($n = 6$). **f, g** Histology of hMSC pellets **f** treated with the indicated drugs or **g** infected with the indicated shRNA lentiviruses. Scale bars: 100 μm. **h** Knockdown efficiency of sh*TNKS* and sh*TNKS2* in hMSC ($n = 4$). **i–l** Tankyrase knockdown in hMSCs regenerates articular cartilage in vivo. **i** Gross appearance (top; scale bar: 500 μm), Alcian blue staining (middle), and human mitochondria immunostaining (bottom; scale bar: 25 μm) of cartilage lesions. ▼ indicates the graft sites. Cartilage regeneration as evaluated by using the **j** ICRS macroscopic score system ($n = 6$) and immunostaining of **k** tankyrase, **l** SOX9, cartilage matrix proteins, and β-catenin in repair tissues in the defects. Scale bar in **k**: 25 μm. Scale bars in **l**: 20 μm. **a, d, e, h, j** Data represent means ± s.e.m. $^*P < 0.05$, $^{**}P < 0.01$, $^{***}P < 0.001$ by ANOVA (**a, d, e, j**) or *t* test (**h**). **i, j, k, l** The relative chromogen intensity or the percentage of immunopositive cells is indicated

responsible for the recognition and ubiquitination of various PARylated substrates, including Axin[33], PTEN[69], and AMOTs[71]. RNF146 forms a complex with tankyrase and directly interacts with PARylated Axin through its WWE domain, thereby promoting its degradation[33,34]. Although the knockdown of RNF146 effectively reduced β-catenin stability and activity, it affected neither SOX9 stability and activity nor cartilage matrix gene expression. Our findings suggest the potential for the presence of another PAR-dependent E3 ligase responsible for tankyrase-mediated SOX9 regulation. Our analysis of endogenous tankyrase interactomes reveals that thyroid hormone receptor interactor 12 (TRIP12) and HECT, UBA, and WWE domain-containing 1 (HUWE1) are the additional E3 ubiquitin ligases that form complexes with tankyrase in chondrocytes. Intriguingly, both TRIP12 and HUWE1 contain the WWE domain that is predicted to mediate interactions with PARylated substrates[33,80,81].

Although we demonstrated that tankyrase-mediated PARylation of SOX9 plays an essential role in regulating SOX9 ubiquitination and degradation, inhibiting tankyrase activity or mutating the tankyrase-binding site in SOX9 did not completely abolish the PARylation of SOX9. We speculate that the residual PARylation of SOX9 may involve modifications mediated by other PARP family members. Extensive molecular-level investigation is warranted to illustrate the role of other forms of SOX9 PARylation that are not mediated by tankyrase.

Taken together, our results have direct implications for the development of regenerative therapy for osteoarthritic cartilage lacking its load-bearing function as a result of cartilage matrix loss. We demonstrate that tankyrase inhibition stimulates cartilage-specific anabolism, illustrating an intriguing possibility for the reconstruction of cartilage at both molecular and organismal levels. The reparative capacity of tankyrase inhibition may be attributable to the combinatorial effects that invert OA-associated gene expression landscape in chondrocytes and stimulate chondrogenesis of MSCs. Our findings may provide fundamental insights into the role of tankyrase-mediated PARylation in fine-tuning the activity of transcription factors associated with development processes and disease pathogenesis. We expect our findings to provide a guideline for future OA therapeutic strategies circumventing surgical intervention and aiming at cartilage regeneration.

## Methods

**In silico analysis of multi-tissue BXD transcriptomes**. Cartilage (GN208)[25], bone femur (GN411)[82], kidney (GN118), lung (GN160)[83], and brain (GN123)[84] datasets were obtained from GeneNetwork (www.genenetwork.org). illumina-Mousev1.db (http://bioconductor.org/packages/illuminaMousev1.db/) and illumi-naMousev1p1.db (http://bioconductor.org/packages/illumina Mousev1p1.db/) were used for probe annotation in the cartilage and bone femur datasets, respectively. These two databases provided information on the quality grade of each probe, and whether a probe overlaps with any known single-nucleotide polymorphisms (SNPs). A probe that did not overlap with any known SNPs, perfectly and uniquely matched the target transcript, and also had the highest expression used for each transcript. The processed cartilage dataset with the expression profile

of 16,074 genes across 16 BXD mouse strains was used to calculate the Pearson's correlation coefficients between gene expression levels. The resulting correlation matrix was hierarchically clustered by using fastcluster package (https://cran.r-project.org/package=fastcluster) in R (complete linkage and Euclidean distance). For the data presented in Supplementary Fig. 1c, principal component analysis was performed by using the R stats package. For the kidney, lung, and brain datasets, we used the mouse4302.db (http://bioconductor.org/packages/mouse4302.db/) for probe annotation, and the probe with the highest expression was used for each transcript. For the data presented in Fig. 1a and Supplementary Fig. 1a, gene co-expression matrices were clustered by using the hierarchical clustering algorithm (complete linkage and uncentered correlation distance) in Cluster (http://bonsai.hgc.jp/~mdehoon/software/cluster/software.htm), and correlation heatmaps were drawn with Gitools[85]. For the data presented in Fig. 1b, factor analysis was performed by using IBM SPSS Statistics (www.ibm.com/analytics/us/en/technology/spss/).

**Primary culture of mouse articular chondrocytes**. For the primary culture of mouse articular chondrocytes, cells were isolated from femoral condyles and tibial plateaus of 4–5-day-old ICR mice by 0.2% collagenase digestion[86]. Chondrocytes were maintained in DMEM supplemented with 10% fetal bovine serum (FBS), 100 units/ml penicillin, and 100 μg/ml streptomycin, and cells were treated as indicated in each experiment. Transfection was performed with METAFECTENE PRO (Biontex) according to the manufacturer's protocol. Small interfering RNAs (siR-NAs) used for RNA interference (RNAi) in mouse articular chondrocytes are listed in Supplementary Table 2. All siRNAs, including negative control siRNA, were purchased from Bioneer. Recombinant mouse Wnt-3a (315-20) was purchased from PeproTech, and recombinant mouse Dkk-1 (5897-DK) was purchased from R&D Systems.

**Reverse transcription polymerase chain reaction and quantitative PCR**. Total RNAs were extracted by using TRI reagent (Molecular Research Center, Inc.). RNAs were reverse transcribed by using EasyScript Reverse Transcriptase (Transgen Biotech). Then, cDNA was amplified by PCR or qPCR with the primers listed in Supplementary Table 3. qPCR was performed with SYBR TOPreal qPCR 2× preMIX (Enzynomics) to determine transcript abundance. Transcript quantity was calculated by using the $\Delta\Delta C_t$ method, and *Hprt* or *HPRT1* levels were used as housekeeping controls.

**Whole-cell lysate preparation**. Whole-cell lysates were prepared in radio-immunoprecipitation assay (RIPA) buffer (150 mM NaCl, 1% NP-40, 50 mM Tris, pH 8.0, 0.5% sodium deoxycholate, and 0.1% sodium dodecyl sulfate (SDS)) supplemented with a protease inhibitor cocktail (Sigma-Aldrich). The lysates were quantified by using a BCA assay and analyzed by SDS polyacrylamide gel electrophoresis (PAGE). Protein band intensity was quantified by densitometric analysis by using ImageJ[87].

**Antibodies**. Anti-FLAG tag antibody (3165) was purchased from Sigma-Aldrich. Antibodies against green fluorescent protein (GFP) (sc-9996), Sox9 (sc-20095), Sox9 (sc-166505), Tankyrase-1/2 (sc-8337), Tankyrase-1/2 (sc-365897), Ubiquitin (sc-8017), Actin (sc-1615), α-Tubulin (sc-23948), and Histone H3 (sc-517576) were obtained from Santa Cruz Biotechnology. Sox9 (sc-20095) antibody was used only in Fig. 3e, n and Sox9 (sc-166505) antibody was used only in Fig. 3f. Tankyrase-1/2 (sc-365897) antibody was used only in Fig. 6a, b, 7k and Supplementary Fig. 4. Antibodies against aggrecan (AB1031), type II collagen (MAB8887), and human mitochondria (MAB1273) were purchased from Millipore, and antibodies against Myc tag (2276) and Sox9 (82630) were purchased from Cell Signaling Technology. Prior to detection of aggrecan, samples were treated with chondroitinase ABC (C3667) from Sigma-Aldrich. Antibodies against HA tag (ab9110) and MMP13 (ab51072) were purchased from Abcam. Anti-β-catenin antibody (610154) was obtained from BD Biosciences. Anti-Poly(ADP-ribose) antibody (AG-20T-0001) was purchased from AdipoGen and was used only in Fig. 3l. Anti-

Poly(ADP-ribose) antibody (4335-AMC) was purchased from Trevigen. All antibodies were used according to the manufacturer's protocol.

**Chemical Inhibitors**. XAV939 (X3004), IWR-1 (I0161), JW55 (SML0630), WIKI4 (SML0760), IWP-2 (I0536), and cycloheximide (C7698) were obtained from Sigma-Aldrich. G007-LK (B5830) and MG132 (A2585) were purchased from Apexbio, G244-LM (1563007-08-8) was from AOBIOUS, MN-64 (HY9351) from MedChem Express, AZ6102 (S7767) from SelleckChem, and TC-E 5001 (5049) from Tocris. Tankyrase inhibitors were classified into three different classes depending on their mode of action[24,88]. ABT-888 (11505) was purchased from Cayman, and ADP–HPD (118415) was purchased from Calbiochem.

**RNA sequencing (RNA-seq)**. Primary cultured mouse articular chondrocytes were treated with DMSO or 10 μM of XAV939 or IWR-1 for 108 h or transfected with control siRNA or *Tnks* and *Tnks2* siRNAs. Three biological replicates were used for each group. One microgram of high-quality RNA samples (RIN > 7.0) were used to construct RNA-seq libraries with the TruSeq Stranded mRNA Library Prep kit (Illumina). Libraries were validated with an Agilent 2100 Bioanalyzer. RNA-seq was performed on an Illumina HiSeq 2500 sequencer at Macrogen. The sequence reads were trimmed with Trimmomatic[89] and mapped against the mouse reference genome (mm10) by using TopHat[90]. Read counts per gene were calculated by using HTSeq[91]. Differential expression analysis was conducted by using the DESeq2 R package[92]. DEGs were selected by using a |fold change| cutoff of >3 and a FDR $q$ cutoff of $<1 \times 10^{-5}$. DEGs in at least one condition were clustered with hierarchical clustering algorithm (ward.D linkage with euclidean distance) by using gplots R package. GO analysis was conducted by using Enrichr[93]. Heatmaps of DEGs that are in the cartilage-signature gene set or the OA-signature gene sets were drawn with Gitools. Venn diagram for the upregulated or downregulated DEGs of the three different tankyrase inhibition groups (si*Tnks* + si*Tnks2* vs. siControl, XAV939 vs. DMSO, and IWR-1 vs. DMSO) was drawn by using eulerAPE[94].

**GSEA analysis**. Genes were ranked according to the shrunken log$_2$ fold change calculated via DESeq2. GSEA[95] was performed in pre-ranked mode, with all default parameters, for the cartilage-signature gene set or the OA-signature gene sets. A thousand permutations were used to calculate $P$ values.

**Generation of a cartilage-signature gene set**. Microarray data for nasal chondrocytes at embryonic day 17.5 and rib chondrocytes at postnatal day 1 were obtained from GSE69108[18]. Microarray data for mouse embryonic fibroblasts (MEFs) were obtained from GSM577694, GSM577695, and GSM577696 of GSE23547[96]. The limma R package[97] was used to compute differential expression between nasal chondrocytes and MEFs or between rib chondrocytes and MEFs. The probe with the highest expression was used for each transcript. Genes with a fold change of >5 and a FDR $q$ of $<1 \times 10^{-5}$ in both nasal chondrocytes and rib chondrocytes compared with MEFs were selected as cartilage-signature genes. The cartilage-signature genes are listed in Supplementary Table 9.

**Subcellular fractionation**. Cells were lysed with hypotonic buffer (20 mM Tris-HCl, pH 7.4, 10 mM NaCl, and 3 mM MgCl$_2$) supplemented with the protease inhibitor cocktail. After incubation in ice, 10% NP-40 was added to the cell lysate. The homogenate was centrifuged at $720 \times g$ for 10 min, and the supernatant was collected as the cytoplasmic fraction. The pellet was washed twice with hypotonic buffer and resuspended in cell extraction buffer (10 mM Tris, pH 7.4, 2 mM Na$_3$VO$_4$, 100 mM NaCl, 1% Triton X-100, 1 mM EDTA, 10% glycerol, 1 mM EGTA, 0.1% SDS, 1 mM NaF, 0.5% deoxycholate, and 20 mM Na$_4$P$_2$O$_7$) supplemented with the protease inhibitor cocktail. The resuspended pellet was centrifuged at $14,000 \times g$ for 30 min, and the supernatant was kept as the nuclear fraction. The cytoplasmic and nuclear fractions were quantified by using the BCA assay. Then 10 μg of proteins per fraction sample were loaded and analyzed by SDS-PAGE. Protein band intensity was quantified by densitometric analysis by using ImageJ.

**Lentiviral infection and selection of HEK293 cells**. psPAX2, pMD2.G, and pLVX-Puro-TK-FLAG-*SOX9*-WT or pLVX-Puro-TK-FLAG-*SOX9*-ΔTBD1/2 were transfected into HEK293T cells. After 2 days, cell supernatants were harvested and filtered through a 0.45-μm filter. HEK293 cells were treated with 8 μg/ml polybrene and transduced with the indicated lentiviruses. HEK293-FLAG-SOX9-WT and HEK293-FLAG-SOX9-ΔTBD1/2 were then selected by using 1 μg/ml puromycin for 2 days.

**Immunoprecipitation**. Cell lysates were prepared by using EBC200 buffer (50 mM Tris-HCl, pH 7.4, 150 mM NaCl, 0.5% NP-40, and 1 mM EDTA) supplemented with the protease inhibitor cocktail. Only for Fig. 3m, q, cell lysates were prepared by using RIPA buffer supplemented with the protease inhibitor cocktail and diluted with EBC200 buffer supplemented with the protease inhibitor cocktail. Cell lysates were used for pull-down with the indicated antibodies and protein A/G-Sepharose beads (GE Healthcare), anti-FLAG M2 magnetic beads (M8823; Sigma-Aldrich), or

Anti-Ub Agarose-TUBE2 (UM402; LifeSensors). The bound proteins were subjected to SDS-PAGE or LC–MS/MS analysis.

**Endogenous tankyrase pulldown and mass spectrometry**. Primary cultured mouse articular chondrocytes were grown for 4 days. Cells were lysed, and lysates were incubated with normal rabbit IgG or anti-tankyrase antibody. The bound proteins were eluted with 8 M urea in 50 mM NH$_4$HCO$_3$ buffer, pH 8.2 for 1 h at 37 °C, and in-solution digestion was performed[98]. Peptide sequencing was carried out by LC–MS/MS on a Thermo Ultimate 3000 RSLCnano high-pressure liquid chromatography system coupled to a Thermo Q-Exactive Hybrid Quadrupole-Orbitrap mass spectrometer. LC–MS/MS raw data were converted into .mzML files by using ProteoWizard MSConvert[99], and the MS-GF+ algorithm[100] with a parameter file consisting of no enzyme criteria and static cysteine modification ( + 57.022 Da) was used for comparison of all MS/MS spectra against the mouse Uniprot database. The final peptide identifications had <1% false-discovery rate (FDR) $q$, at the unique peptide level. Only fully tryptic and semitryptic peptides were considered. For each biological replicate, proteins that were detected only once and proteins that were co-immunoprecipitated with normal rabbit IgG were not considered. For proteins detected in more than one biological replicate, the peptides and proteins are listed in Supplementary Data 1. The Venn diagram was drawn with eulerAPE.

**In silico prediction of tankyrase substrate proteins**. The $8 \times 20$ position-specific scoring matrix generated in Guettler et al.[30] was used to calculate a TTS for each octapeptide in the proteins identified by LC–MS/MS.

$$\text{TTS} = \frac{\sum_{\text{pos.}=0}^{8} \text{PSSM}_{\text{pos.}}}{\max\left(\sum_{\text{pos.}=0}^{8} \text{PSSM}_{\text{pos.}}\right)}.$$

Only those proteins having at least one octapeptide with a TTS of ≥0.385 were considered. This cutoff is the TTS of the tankyrase-binding motifs of mouse AXIN1 and AXIN2. AXIN1 and AXIN2, the known tankyrase substrates[27], have the lowest maximum TTS among the known tankyrase substrates, due to the suboptimal amino acids at the fourth and fifth positions[30]. For further screening, the *chondrogenesis* category in IPA (https://www.qiagenbioinformatics.com/products/ingenuity-pathway-analysis/) was used. The mouse proteins in the IPA *chondrogenesis* category are listed in Supplementary Table 7. For the candidate proteins, IUPred disorder scores were calculated for the octapeptides with a TTS of ≥0.385. The heatmap of TTS and IUPred disorder scores for candidate proteins was drawn with Gitools.

**Cell line culture**. HEK293 and HEK293T cells were cultured in DMEM containing 10% FBS, 100 units/ml penicillin, and 100 μg/ml streptomycin. Transfection was performed with METAFECTENE PRO (Biontex) or PEI transfection reagent (Sigma-Aldrich) according to the manufacturer's protocol. The siRNAs used in HEK293T are listed in Supplementary Table 2. The siRNA sequences targeting *TNKS* or *TNKS2* were described previously[27].

**Plasmids**. Human *SOX9* cDNA (hMU008919) was purchased from Korea Human Gene Bank and subcloned into a pcDNA3-HA plasmid or a p3xFLAG-CMV10 plasmid. To express human SOX9 under TK promoter, *Renilla* luciferase gene in a pRL-TK plasmid was replaced by 3xFLAG-*SOX9*. TK-3xFLAG-*SOX9* was subcloned into a pLVX-Puro plasmid. To generate mutant constructs, PCR-mediated mutagenesis was conducted. The GFP-tagged human TNKS plasmid was a gift from Dr. Chang-Woo Lee[101], and the Myc-tagged human TNKS2 plasmid was a gift from Dr. Junjie Chen[69,71]. The FLAG-tagged human TNKS2 plasmid and the FLAG-tagged human TNKS2 M1054V plasmid were gifts from Dr. Nai-Wen Chi[102]. The 4 × 48-p89 SOX9-dependent *Col2a1* luciferase reporter construct was a gift from Dr. Veronique Lefebvre[36]. Human *TNKS2* cDNA was subcloned into a pEGFP-C1 plasmid to construct a GFP-tagged human TNKS2 plasmid. A control shRNA sequence was inserted into the pLKO.1 puro and pLKO.1 hygro plasmids. Human *TNKS* and *TNKS2* shRNA sequences were inserted into the pLKO.1 puro and pLKO.1 hygro plasmids, respectively. The shRNA sequences targeting human *TNKS* or *TNKS2* were as described previously[27]. Mouse *Tnks* and *Tnks2* shRNA sequences were inserted into the pLKO.1 puro and pLKO.1 hygro plasmids, respectively. Mouse *Rnf146* shRNA sequence was inserted into the pLKO.1 puro plasmid. The shRNA sequence targeting mouse *Tnks* was as described previously[68]. The primers used to generate the above plasmids are listed in Supplementary Tables 4–6.

**In situ PLA**. Primary cultured mouse articular chondrocytes were used for in situ PLA. Duolink® PLA was performed according to the manufacturer's protocol (Sigma-Aldrich). Antibodies against Sox9 (sc-166505) and Tankyrase-1/2 (sc-8337) were used to recognize endogenous mouse SOX9 and endogenous mouse tankyrase, respectively.

**Sequence alignment of SOX9–TBD1 and TBD2 among vertebrates**. For the sequence alignment of TBD1 and TBD2 of SOX9 among vertebrates, NP_000337.1

(*Homo sapiens* SOX9), NP_035578.3 (*Mus musculus* SOX9), NP_989612.1 (*Gallus gallus* SOX9), NP_001016853.1 (*Xenopus tropicalis* SOX9), and NP_571718.1 (*Danio rerio* SOX9) were used.

**Structural modeling of protein–peptide interactions**. GalaxyPepDock[103] was used for modeling of the ARC4 domain of human TNKS2 in complex with the TBD1 or TBD2 peptide of human SOX9. The structures of ARC4:3BP2 (PDB ID: 3TWR) and ARC4:MCL1 (PDB ID: 3TWU) were obtained from Guettler et al.[30]. The ARC4 domain of human TNKS2 (PDB ID: 3TWU_A) and MCL1 peptide (PDB ID: 3TWU_B) were used as templates. The MCL1 peptide was substituted by the TBD1 (255–266 aa) or TBD2 (269–280 aa) peptide of human SOX9 and docked into a complex. The best predicted model for each of ARC4:SOX9–TBD1 and ARC4:SOX9 TBD2 was selected. The model structures were superimposed with ARC4:3BP2 and ARC4:MCL1 and visualized by using the BIOVIA Discovery Studio Visualizer (http://accelrys.com/products/collaborative-science/biovia-discovery-studio/visualization.html).

**Cycloheximide chase analysis**. HEK293 cells were treated with 50 μM of cycloheximide for the indicated number of hours before lysis. Protein samples were subjected to SDS-PAGE to analyze protein stability.

**Reporter gene assay**. A firefly luciferase reporter plasmid with SOX9-dependent *Col2a1* enhancer elements was used to quantify the transcriptional activity of SOX9. To quantify β-catenin transcriptional activity, the TOPFlash reporter plasmid[104] was used. Primary mouse articular chondrocytes or HEK293T cells were transfected with both a reporter plasmid and a constitutive *Renilla* luciferase plasmid. Cells were also treated with siRNAs or drugs as indicated. *Renilla* and firefly luciferase activity were sequentially measured by using a Dual Luciferase Assay Kit (Promega). *Renilla* luciferase was used as a control.

**List of SOX9 target genes in chondrocytes**. Based on Oh et al.[37], genes with a $\log_2$(fold change) of <−2 after *Sox9* deletion in mouse rib chondrocytes and associated with SOX9 ChIP-Seq peaks in mouse rib chondrocytes were selected as SOX9 target genes in chondrocytes. The SOX9 target genes in chondrocytes are listed in Supplementary Table 8.

**Generation of OA-signature gene sets**. Based on Dunn et al.[41], genes with a |fold change| of >2 and a FDR $q$ of <$1 \times 10^{-5}$ in damaged sites of articular cartilage compared with intact sites within the same patients with OA were selected, and converted into mouse nomenclature by using the biomaRt R package[105]. Genes that are upregulated and downregulated in osteoarthritic cartilage are listed in Supplementary Tables 10 and 11, respectively.

**Preparation and in vivo evaluation of hydrogels**. 6-O-Palmitoyl-l-ascorbic acid (76183) was purchased from Sigma-Aldrich. Hydrogels (final concentration: 4% m/v) were prepared by dissolving 6-O-Palmitoyl-l-ascorbic acid with or without drugs in DMSO and mixing with sterilized distilled water[42]. DiD percholate (5702) purchased from Tocris was loaded into the hydrogels and used for imaging of controlled release in mouse knee joints. PBS-suspended hydrogel (10 μl, PBS: hydrogel = 1:1) containing 50 pmol DiD was administered intra-articularly, and at 1–9 days post injection, light-emitting diode and fluorescence images of knee joints were obtained. LuminoGraph II (Atto) was used to acquire the images.

**Experimental OA in mice**. Eight-week-old male ICR mice were used for experimental OA. Post-traumatic OA was induced by DMM surgery; sham operation was used as a control[106]. In all, 10 μl of PBS-suspended hydrogel (PBS:hydrogel = 1:1) containing vehicle or 10 nmol drugs was administered intra-articularly. To generate the data presented in Supplementary Fig. 4, experimental OA was induced by DMM surgery in 12-week-old male C57BL/6 mice and their knee joints were processed for histological and immunostaining analysis at 2 and 8 weeks after surgery.

**Human OA cartilage**. Human OA cartilage specimens were sourced from OA patients undergoing total knee replacement at SNU Boramae Medical Center. The Boramae Medical Center Institutional Review Board (IRB No. 30-2017-48) approved the collection of the human specimens and the Seoul National University Institutional Review Board (IRB No. E1803/003-009) approved the use of these specimens. Written informed consent was obtained from all participants before the total knee replacement arthroplasty operative procedure.

**Histology and immunohistochemistry**. Mouse and rat knee joint samples and human cartilage samples were fixed with 4% paraformaldehyde overnight at 4 °C. Samples used in Supplementary Fig. 4 were decalcified in 0.5 M EDTA, pH 7.4, for 2–4 weeks at 4 °C. Samples used in Supplementary Fig. 5e–i, Fig. 7i–l, and Supplementary Figs. 6c–f, 8 were decalcified with 8% nitric acid for 8 h at room temperature. The human cartilage samples were not decalcified. All samples were embedded in paraffin. Mouse and rat paraffin blocks were sectioned at a thickness

of 6 μm, and human paraffin blocks were sectioned to a thickness of 5 μm. For Safranin O staining, Alcian blue/Fast red staining, or immunostaining, sections were deparaffinized in xylene and hydrated by using a graded ethanol series. For immunohistochemistry, type II collagen antibody was used at 1:100–1:200 dilutions, and the other primary antibodies were used at a 1:50 dilution. All mouse histology images were acquired from medial tibial plateau except β-catenin immunostaining images where medial femoral condyle was used for imaging. To assess cartilage destruction in DMM mouse model, Safranin O stained samples were graded based on the Osteoarthritis Research Society International (OARSI)[107] by three blinded observers. On the basis of OARSI grading system, we primarily conducted integrative evaluation focusing on structural changes and proteoglycan loss in articular cartilage as a measure of cartilage destruction. Cartilage regeneration in osteochondral defect model was scored according to the International Cartilage Repair Society (ICRS) scoring system[108,109] by three blinded observers. Image-Pro Premier (http://www.mediacy.com/support/imagepropremier) was used for the quantification of immunostaining images. Signal-positive regions or cells were defined based on the chosen threshold. For intracellular proteins, the number of immunopositive cells or the percentage of immunopositive cells were calculated. For extracellular proteins, the relative chromogen intensity was calculated.

**Mouse limb-bud micromass culture**. For the micromass culture of mesenchymal cells, limb-bud cells were isolated from E11.5 ICR mouse embryos. In total, $2.0 \times 10^7$ cells/ml were suspended in DMEM supplemented with 10% FBS, 100 units/ml penicillin, and 100 μg/ml streptomycin, and 15-μl drops were spotted on culture dishes. After 24 h, cells were treated as indicated for 3 days and subjected to Alcian blue staining or RNA extraction. For Fig. 7a, cells were harvested at days 1, 3, and 5 and subjected to RNA extraction.

**Chondrogenic differentiation of hMSCs**. hMSCs were purchased from Lonza and Thermo Scientific. hMSCs were cultured in α-MEM supplemented with 20% FBS, 100 units/ml penicillin, 100 μg/ml streptomycin, and 250 ng/ml amphotericin B. To induce chondrogenesis, $2.5 \times 10^5$ hMSCs were centrifuged to form a pellet in α-MEM supplemented with 20% FBS, 100 units/ml penicillin, 100 μg/ml streptomycin, and 250 ng/ml amphotericin B. After 3 days, the medium was changed to chondrogenic medium consisting of DMEM/F-12 supplemented with 100 units/ml penicillin, 100 μg/ml streptomycin, 250 ng/ml amphotericin B, 1.25 mg/ml BSA, 1% insulin–transferrin–selenium, 1 mM sodium pyruvate, 50 μM L-aspartic acid, 50 μM L-proline, 100 nM dexamethasone, and 10 ng/ml of TGF-β1 with or without indicated drugs. On day 21 (for drug treatment) or day 28 (for siRNA treatment), cells were harvested and subjected to Alcian blue/Fast red staining. RNA-Seq data of human MSCs (GSE109503[47]) during chondrogenesis at days 0, 1, 3, 7, 14, and 21 were obtained from gene expression omnibus repository at the National Center for Biotechnology Information (NCBI). Differential expression analysis was conducted by using the DESeq2 R package.

**shRNA-mediated knockdown in hMSCs**. psPAX2, pMD2.G, and the indicated shRNAs were transfected to HEK293T cells. After 3 days, cell supernatants were harvested and filtered through a 0.45-μm filter. hMSCs were treated with 8 μg/ml polybrene and infected with the indicated lentiviruses. Twenty-four hours after infection, hMSCs were selected with 5 μg/ml puromycin and 200 μg/ml hygromycin for 4 days.

**Rat osteochondral defect model**. Twelve-week-old male Sprague Dawley rats were used for an osteochondral defect model. To expose the articular cartilage in the knee joints, a medial parapatellar incision was made, and the patella was slightly displaced toward the medial condyle. A full-thickness cartilage defect (3 mm × 1 mm × 1 mm) was created by using a 1-mm-diameter spherical drill at the surface of the femoral patellar groove. At the same time, hMSCs were suspended in 10 μl of fibrin glue (TISSEEL) by tapping and implanted on the defect. To avoid immune rejection, cyclosporine A (C988900) from Toronto Research Chemicals was injected intraperitoneally every day. At 8 weeks, rats were killed for histological analyses.

**Statistics**. All experiments were carried out independently at least three times. All images are representative of at least three independent trials. For parametric tests, two-tailed Student's $t$ test or one-way analysis of variance followed by Fisher's least significant difference post hoc test was used. For nonparametric tests, Mann–Whitney test or Kruskal–Wallis test followed by Mann–Whitney test were used. All statistical analysis was performed by using IBM SPSS Statistics. A $P$ value < 0.05 was considered statistically significant. Cell culture samples and animals were randomly assigned to each group with about equivalent numbers in each group, and all samples were analyzed in a blinded manner. The sample sizes were determined on the basis of previous study[50], and no exclusion criteria were applied.

**Study approval**. All animal studies were conducted with the approval of the Seoul National University Institutional Animal Care and Use Committee (IACUC). We conformed to the ARRIVE guidelines for reporting animal experiments[110].

**Reporting summary**. Further information on research design is available in the Nature Research Reporting Summary linked to this article.

## Data availability
RNA-seq datasets are deposited in the Gene Expression Omnibus (GSE137899).

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

## Acknowledgements

We thank the staff from the Department of Orthopedic Surgery, SNU College of Medicine, Boramae Hospital, and the patients for participating in the study. We would also like to acknowledge Jong Min Lee (Dongguk University Ilsan Hospital) for technical assistance on osteochondral defect model. We thank Dong Hoon Kang (Ewha Womans University) for helpful advice on PARylation and ubiquitination assays. We also thank all group members of J.-H.K. lab for helpful discussions. This work was supported by grants from the National Research Foundation of Korea (NRF-2015M3A9E6028674, NRF-2016R1A5A1010764, and NRF-2017M3A9D8064193), Korean Ministry for Health and Welfare (1520070), the Institute for Basic Science from the Ministry of Science, ICT, and Future Planning of Korea (IBS-R008-D1), Creative-Pioneering Researchers Program through SNU, and Suh Kyungbae Foundation.

## Author contributions

S.K., S.H., and J.-H.K. designed the study. S.K. and S.H. performed the most in vitro and in vivo experiments. Y.K., H.-S.K., Y.-R.G., D.K., Y.C., J.L., and Y.S. conducted the cell-

based assays. Y.K., H.K., and J.L. conducted the immunohistochemistry analysis. H.-S.K. performed the mass spectrometry analysis. M.J.C., C.B.C., and S.-B.K. collected and inspected the human patient samples. S.K., S.H., Y.K., H.-S.K., Y.-R.G., D.K., Y.C., H.K., J.L., and J.-H.K. analyzed the data. S.K., S.H., and J.-H.K. wrote the paper. All authors read and edited the paper. J.-H.K. supervised the study.

## Competing interests

S.K., S.H., Y.C., and J.-H.K. are the inventor authors on pending patent application, "Use of TNKS inhibitors for regeneration of cartilage", Korea patent application no. 10-2018-0139120. The remaining authors declare no competing interests.
