## [Peer Review File · Nature Communications]

Reviewers' Comments:

Reviewer #1:

Remarks to the Author:

In this study, Han et. al. examined the function of TNKS in cartilage matrix anabolism. Authors first identified TNKS as a hit from quantitative genetic analysis of articular cartilage in BXD mouse reference population. Authors next demonstrated that inhibition of TNKS induces cartilage matrix gene expression, and this activity of TNKS is mediated by SOX9, but not beta-catenin. Authors further showed that TNKS physically interacts with SOX9 and promotes degradation of SOX9. Although the study is interesting, there are several major holes in this story. Authors have not provided convincing evidence the activity of TNKS is mediated by SOX9 but not beta-catenin. This is critical because the role of beta-catenin signaling in chondrocyte differentiation and chondrogenic gene expression is established, and TNKS inhibitor inhibits beta-catenin signaling. In addition, authors have failed to provide convincing data that TNKS promotes PARsylation and degradation of SOX9.

Major points:

1. Authors did not provide convincing evidence that the effect of TNKS inhibitor is not mediated by beta-catenin inhibition. Wnt/beta-catenin signaling is known to inhibit chondrocyte differentiation and chondrogenic gene expression. Wnt inhibitor SM04690 is showing promise as a disease modifying drug for OA (PMID: 28711582). Porcupine inhibitor and TNKS inhibitor, which inhibit Wnt signaling through distinct mechanisms, showed activities in animal model of OA (PMID: 29415892). In principle, the effect of TNKS inhibitor could be mediated by Wnt inhibition. Authors showed that beta-catenin knockdown did not induce cartilage matrix gene expression (Fig.2e), and concluded the effect of TNKS inhibitor is not mediated by beta-catenin inhibition. However, this negative result could result from loss of membrane function of beta-catenin. Another piece of evidence authors provided is that treatment of iCRT14 did not increase cartilage matrix gene expression (Fig. 2g). iCRT14 is not a Wnt inhibitor with clean mechanism of action, so it is an ideal reagent to answer this question. Also, cells were treated with iCRT14 only for 24 hr. In Fig. 1f and 1g, cells were treated with TNKS inhibitor for 108 hr. Authors should test whether treatment of cells with porcupine inhibitor, which blocks both canonical and noncanonical Wnt signaling, and DKK1, which only blocks canonical Wnt signaling, affects cartilage matrix gene expression.
2. Authors did not provide convincing evidence that SOX9 is a substrate of TNKS. Figure 3l supposedly showed that PARsylation of Sox9 is diminished when TBD1/2 were deleted. However, the effect is marginal. Also, there is no description of the experimental details. Is this a GST-WWE pulldown experiment? Authors should also test the effect of TNKS inhibitor on PARsylation of SOX9.
3. Authors failed to show convincing data that TNKS regulates protein stability of SOX9. In Fig. 3m and 3n, authors showed that protein levels of SOX9 were increased upon TNKS inhibition. However, authors did not show whether mRNA levels of SOX9 were also affected upon TNKS inhibition. TNKS TBD1/2 deletion mutant appears to be degraded more slowly in a CHX experiment (Fig. 3o). However, wildtype SOX9 and SOX9 TBD1/2 deletion mutant were expressed at similar levels at 0 hr, which is not consistent with authors' hypothesis. The protein level of SOX9 TBD 1/2 deletion mutant should be much higher than that of wildtype SOX9 if these two proteins are expressed using a weak promoter. Authors also did not test the effect of TNKS inhibitor on ubiquitination of SOX9. With this in mind, it is interesting that RNF146 does not regulate the protein level of SOX9, since all proteins known to be degraded TNKS are ubiquitinated by RNF146. Much more rigorous studies are needed to demonstrate the effect of TNKS inhibitor on ubiquitination and degradation of SOX9. For example, authors should perform S35 based pulse chase study of endogenous SOX9 with or without treatment of TNKS inhibitor.
4. SOX9 is a master regulator of chondrogenesis. The finding that knockdown of SOX9 abolished the effect of TNKS inhibitor on cartilage matrix gene expression (Fig. 5i) does not prove the functional interaction between SOX9-TNKS. The only evidence shown in Fig 5h is based on overexpression and it is somewhat weak. A much better experiment would be in vivo knock in.

Reviewer #2:

Remarks to the Author:

Han S et al. reports that tankyrase accelerates OA progression through SOX9 PARylation and the subsequent decrease of cartilage anabolism. They identified tankyrase as a regulator of cartilage anabolic axis by in silico analyses. Expression of Tnks and Tnks2 is inversely correlated with that of Col2a1 and Acan. siRNA against Tnks&Tnks2 or tankyrase inhibitors increase expression of cartilage matrix genes. They further identified SOX9 as a target of tankyrase. Protein-protein binding between SOX9 and tankylase is shown by in situ proximity ligation assay and co-IP. Tankyrase inhibitor enhances SOX9 protein level and expression of SOX9-target genes. Intra-articular injection of tankyrase inhibitor ameliorates OA development in DMM mice. Tankyrase inhibition stimulates chondrogenesis of MSC. The present findings are novel and interesting. Meanwhile, several points should be addressed.

Major points

1. The authors show that tankyrase inhibition enhanced cartilage anabolism in vitro and in vivo. However, they did not show how tankyrases regulate chondrocytes in OA pathophysiology and chondrogenesis. Is expression of Tnks and Tnks2 increased with OA development? Are they increased or activated in the early stage, or in the late stage? Are they correlated with downregulation of matrix genes in vivo? Are they expressed or activated during chondrogenesis? These analyses are necessary to understand physiological roles of tankyrases in chondrocyte differentiation and articular cartilage homeostasis. These data will be indispensable and much useful for clinical application of tankyrase inhibition in future.
2. They use XAV939 and IWR-1 as tankyrase inhibitors. As they describe, they are widely used as inhibitors of WNT/beta-catenin pathway. XAV939 and IWR-1 enhance beta-catenin degradation through stabilization of Axin. Notably, many previous studies have shown that WNT/beta-catenin is a representative pathway for cartilage degradation, and remarkably suppresses chondrogenesis. Sox9 and WNT/beta-catenin inhibit mutual transactivation (Genes Dev 2004 18:1072, and many following papers). These points should be further analyzed in major experiments in the present study. They show that siCtnnb1 and iCRT did not increase matrix genes (Fig2eg), but these data did not exclude that anabolic effects of XAV939 and IWR-1 are independent of inhibition of WNT/beta-catenin. Fig5j did not exclude the possibility that XAV939 enhanced SOX9 via suppression of WNT/beta-catenin. Considering that tankyrase has many kinds of substrates, it is possible that tankyrase inhibitor exerts anabolic effects in chondrocytes through both SOX9 enhancement and WNT inhibition, as the authors described in Discussion. I believe that multifaceted examinations will make the present study more valuable.
3. Although many data are displayed in Figures, some of them are not well written in the text and the legends. I recommend that the authors describe each result more specifically.
4. All western blot and IHC should be properly quantified.
5. In Fig6, WNT/beta-catenin-related molecules such as beta-catenin or axin should be examined, as well as Col2, Acan, and Sox9. Tnks and tnks2 should be examined in these series Fig6fg, the authors should display which regions were shown. Are they tibial cartilage or femoral, medial or lateral?
6. Fig.7a-c, it is not appropriate to determine chondrogenesis only by alcian blue staining, because it is less specific. Expression of marker genes should be also examined. These experiments are suitable to evaluate SOX9 and WNT/beta-catenin pathway.
7. Fig.7e-h, Tnks, tnks2, and WNT/beta-catenin-related molecules should be examined. Additionally, the authors should demonstrate that the regenerative tissues are derived from hMSC,

e.g., by IHC using human-specific antibodies.

Minor points

8. In the 1st paragraph of Results, in silico analyses performed in Fig1ab should be described more precisely and concretely.

9. I recommend that qPCR and western blot are to be displayed in separate subpanels, for example, Fig.1e,f,g.

10. Please check the figure legends of Fig3f and g.

Reviewer #3:

Remarks to the Author:

General comments to the manuscript:

Using mouse datasets, authors identified tankyrase as a regulator of the cartilage anabolism axis. Tankyrase appears an interesting druggable target since its inhibition leads to SOX9 stabilization with concurrent anabolic effects on articular cartilage. Most likely, SOX9 stabilization is the result of an uncoupling of SOX9 from a poly(ADP-ribosylation) (PARylation)-dependent protein degradation pathway. The authors have done a lot of work. The manuscript is well-written and results are very interesting given the fact that, as mentioned, 'it has been a difficult task [to reconstruct cartilage matrix in osteoarthritic joints] as adult cartilage exhibits marginal repair capacity'.

Major comments:

My main concern relates to the initial analyses and approach taken.

1. Results lines 79-80 'We noted that, among 21 cartilage matrix genes listed up...': why did the authors focus on these 21 genes? That is not completely clear from the Methods section. If a genome wide approach was taken using available microarray gene expression datasets it is a pity to only select these 21 genes to follow-up?

2. Results line 85 'We computed Pearson's correlation coefficients between Factor 1 and various genes...': how many and which genes are these? How were the genes selected?

3. Methods lines 521-523 'For the cartilage and bone femur data sets, a probe having non-overlapping SNPs, 'Perfect' quality, and the highest expression was used for each transcript': what do the authors mean with 'non-overlapping SNPs'? And what is the definition of 'perfect quality'?

Minor comments:

1. Results line 87 '...negative correlations with this anabolic axis' and line 203 '... indicating that tankyrase exhibits a positive correlation with ... catabolic regulators': what about MMP13?

2. Results line 100 'were similarly clustered' please explain, what do you mean?

3. Results line 162 'We explored whether tankyrase inhibition affects the transcriptional activity of SOX9...'. Maybe I missed something, but previously authors showed 'the disruption of the physical interactions between tankyrase and SOX9 ... results in the stabilization of SOX9' (lines 146-147). Isn't it logical that increased SOX9 levels result in the observed increased transcriptional activity of SOX9? Can the authors show a Western blot for the expression levels of SOX9?

4. Results line 171 '...regulation of SOX9 occurs at the protein level rather than at transcription level...'.

5. It is known that loss of tankyrases result in high microsatellite instability due to deregulated poly(ADP-ribosylation)-mediated signaling which might contribute to development of cancers with high microsatellite instability (PMID:21333322). Are the authors not concerned this will affect joint tissues when applying tankyrase inhibitors as a treatment for OA?

6. At some places the logical order of the text could be improved. For example:

a) In line 114 the tankyrase-RNF146 complex is introduced and authors mention 'Consistently,

tankyrase inhibition reduced β -catenin stability...' and not until line 150 the authors explain 'RNF146 is best known to regulate tankyrase-dependent Axin degradation and, hence, β -catenin stabilization'.

b) In line 161 it is mentioned that 'SOX9, a master transcription factor of chondrogenesis, transcribes various cartilage-specific matrix genes'. But SOX9 is introduced already in line 110.

Point-by-point response to the reviewer's comments

Reviewer: 1

General comment. In this study, Han et. al. examined the function of TNKS in cartilage matrix anabolism. Authors first identified TNKS as a hit from quantitative genetic analysis of articular cartilage in BXD mouse reference population. Authors next demonstrated that inhibition of TNKS induces cartilage matrix gene expression, and this activity of TNKS is mediated by SOX9, but not beta-catenin. Authors further showed that TNKS physically interacts with SOX9 and promotes degradation of SOX9. Although the study is interesting, there are several major holes in this story. Authors have not provided convincing evidence the activity of TNKS is mediated by SOX9 but not beta-catenin. This is critical because the role of beta-catenin signaling in chondrocyte differentiation and chondrogenic gene expression is established, and TNKS inhibitor inhibits beta-catenin signaling. In addition, authors have failed to provide convincing data that TNKS promotes PARsylation and degradation of SOX9. Major points:

[Response]

Thank you for your thorough review of our manuscript. We appreciate your constructive comments and suggestions that have helped improve our manuscript immensely. A point-by-point response to your comments is provided below. We have highlighted all the changes in our revised manuscript in blue font.

Major point 1. Authors did not provide convincing evidence that the effect of TNKS inhibitor is not mediated by beta-catenin inhibition. Wnt/beta-catenin signaling is known to inhibit chondrocyte differentiation and chondrogenic gene expression. Wnt inhibitor SM04690 is showing promise as a disease modifying drug for OA (PMID: 28711582). Porcupine inhibitor and TNKS inhibitor, which inhibit Wnt signaling through distinct mechanisms, showed activities in animal model of OA (PMID: 29415892). In principle, the effect of TNKS inhibitor could be mediated by Wnt inhibition. Authors showed that beta-catenin knockdown did not induce cartilage matrix gene expression (Fig. 2e), and concluded the effect of TNKS inhibitor is not mediated by beta-catenin inhibition. However, this

negative result could result from loss of membrane function of beta-catenin. Another piece of evidence authors provided is that treatment of iCRT14 did not increase cartilage matrix gene expression (Fig. 2g). iCRT14 is not a Wnt inhibitor with clean mechanism of action, so it is an ideal reagent to answer this question. Also, cells were treated with iCRT14 only for 24 hr. In Fig. 1f and 1g, cells were treated with TNKS inhibitor for 108 hr. Authors should test whether treatment of cells with porcupine inhibitor, which blocks both canonical and noncanonical Wnt signaling, and DKK1, which only blocks canonical Wnt signaling, affects cartilage matrix gene expression.

[Response]

The reviewer raised a valid concern. As the reviewer mentioned, tankyrase inhibitors are conventionally described as inhibitors of Wnt signaling (Huang *et al*, *Nature*. 2009, 461(7264): 614; Chen *et al*, *Nat. Chem. Biol.* 2009, 5(2): 100), and Wnt/ β -catenin signaling has been shown to suppress the expression of chondrogenic genes (Day *et al*, *Dev. Cell* 2005, 8(5): 739-750; Hwang *et al*, *J. Biol. Chem.* 2004, 279(25): 26597-26604). We tested whether the chondrogenic genes augmented by tankyrase inhibition is dependent on its inhibitory effect on Wnt/ β -catenin signaling. We explored this issue by using Dkk-1, which antagonizes canonical Wnt ligands and IWP-2, a porcupine inhibitor that blocks the secretion of canonical and noncanonical Wnts.

Chondrocytes treated with tankyrase inhibitors or tankyrase-targeting siRNAs augmented the expression of cartilage matrix genes such as *Col2a1* and *Acan* in the absence of exogenous Wnt treatments. We verified whether this result is associated with the effect of Wnt/ β -catenin signaling inhibition. As XAV939 treatment for 72 h induced the mRNA expression of cartilage-specific genes, we treated mouse articular chondrocytes with Dkk-1 or IWP-2 for 72 h. However, the treatment of Dkk-1 or IWP-2 in chondrocytes did not affect the expression of *Col2a1* and *Acan*, suggesting that Wnt/ β -catenin signaling has a negligible effect on cartilage anabolism in our experimental context (i.e., in the absence of exogenous Wnt ligands).

We then conducted our experiments in the presence of exogenous Wnt ligands. Treatment of chondrocytes with Wnt-3a, a canonical Wnt ligand, significantly reduced the expression of *Col2a1* and *Acan* (Response Fig. 1a, b). The addition of recombinant Dkk-1 effectively abolished the Wnt-3a-mediated suppression of these chondrogenic genes. Meanwhile, IWP-2

did not rescue the expression of *Col2a1* and *Acan*, consistent with its inability to block exogenously added Wnt ligands. Therefore, in the presence of abundant Wnt ligands, tankyrase inhibition might exert an additional effect to promote cartilage matrix anabolism in a Wnt/ β -catenin signaling-dependent manner. Nonetheless, our study was mainly conducted in a context where β -catenin signaling plays a marginal role in cartilage matrix gene regulation, suggesting that the tankyrase-PARylation-SOX9 axis can regulate the anabolic pathway independent of Wnt/ β -catenin signaling.

Finally, although iCRT14 has been thought to disrupt the interaction between β -catenin and its transcriptional partner TCF4 (Gonsalves *et al*, *Proc. Natl. Acad. Sci. U.S.A.* 2011, 108(15): 5954-5963), their precise mechanisms to inhibit β -catenin signaling remain unknown, limiting the clear interpretation of our data presented in Fig. 2f, g. Therefore, we replaced the data with new results acquired with authentic Wnt pathway inhibitors, namely, Dkk-1 and IWP-2.

Our new data with the Wnt inhibitors are described in the Results (page 6, 3rd and 4th paragraphs).

Major point 2. Authors did not provide convincing evidence that SOX9 is a substrate of TNKS. Figure 31 supposedly showed that PARylation of Sox9 is diminished when TBD1/2 were deleted. However, the effect is marginal. Also, there is no description of the experimental details. Is this a GST-WWE pulldown experiment? Authors should also test the effect of TNKS inhibitor on PARylation of SOX9.

[Response]

As per the reviewer's suggestion, we evaluated the effect of tankyrase inhibitor (XAV939) on the PARylation of SOX9. HEK293 cells expressing FLAG-tagged human SOX9 were treated with XAV939 or untreated. FLAG-SOX9 was pulled down using anti-FLAG M2 magnetic beads and resolved by SDS-PAGE. The extent of SOX9 PARylation in the presence or absence of XAV939 was detected with anti-PAR antibody along with the negative control that does not express FLAG-SOX9. The PARylation of SOX9 was effectively reduced by tankyrase inhibition (Response Fig. 2). The relevant data are described in the Results (page 8, 1st paragraph).

Interestingly, tankyrase inhibition by XAV939 treatment did not completely abolish the PARylation of SOX9, reminding that SOX9 TBD1/2 deletion mutant had a diminished, but residual PARylation extent (Fig. 3l). We speculate that the residual PARylation of SOX9 might involve the modifications mediated by other PARP family members (Gibson and Kraus, *Nat. Rev. Mol.* 2012, 13(7): 411). Extensive molecular level investigation is warranted to illustrate the role of other forms of SOX9 PARylation that are not mediated by tankyrase. We have now described this issue as a limitation of our study in the Discussion (page 15, 1st paragraph).

Finally, we have provided a detailed description of experimental procedures for information provided in Fig. 3l in the relevant figure legend and Methods of our revised manuscript. Briefly, wild-type or TBD1/2-deleted human SOX9 was tagged with HA epitope and expressed in HEK293T cells. The HA-tagged proteins were pulled down using anti-HA antibodies captured by protein A/G-coupled agarose beads. The immunoprecipitated proteins were resolved on SDS-PAGE and detected using an anti-PAR antibody.

Major point 3. Authors failed to show convincing data that TNKS regulates protein stability of SOX9. In Fig. 3m and 3n, authors showed that protein levels of SOX9 were increased upon TNKS inhibition. However, authors did not show whether mRNA levels of SOX9 were also affected upon TNKS inhibition. TNKS TBD1/2 deletion mutant appears to be degraded more slowly in a CHX experiment (Fig. 3o). However, wildtype SOX9 and SOX9 TBD1/2 deletion mutant were expressed at similar levels at 0 hr, which is not consistent with authors' hypothesis. The protein level of SOX9 TBD 1/2 deletion mutant should be much higher than that of wildtype SOX9 if these two proteins are expressed using a weak promoter. Authors also did not test the effect of TNKS inhibitor on ubiquitination of SOX9. With this in mind, it is interesting that RNF146 does not regulate the protein level of SOX9, since all proteins known to be degraded TNKS are ubiquitinated by RNF146. Much more rigorous studies are needed to demonstrate the effect of TNKS inhibitor on ubiquitination and degradation of SOX9. For example, authors should perform S35 based pulse chase study of endogenous SOX9 with or without treatment of TNKS inhibitor.

[Response]

[Transcriptional regulation of SOX9 by tankyrase inhibition]

We appreciate reviewer's comments. We agree that it is important to examine how the mRNA level of *Sox9* is affected by tankyrase inhibition. During revision, we found that the *Sox9* transcript level was also increased upon simultaneous knock-down of *Tnks* and *Tnks2* in mouse articular chondrocytes although pharmacological inhibition of tankyrase by XAV939 treatment did not result in a statistically significant increase in the mRNA level of *Sox9* (Response Fig. 3a). As we mentioned in our original manuscript, SOX9 is known to bind to its own enhancer and auto-regulate its expression (Mead *et al*, *Nucleic Acids Res.* 2013, 41(8): 4459-4469). Therefore, we speculated that the stabilized SOX9 by tankyrase inhibition might in turn promote the transcription of *Sox9* mRNA. To prove that tankyrase inhibition promotes the SOX9 protein level regardless of transcriptional induction of *Sox9* mRNA, we conducted SOX9-dependent *Col2a1* enhancer-luciferase reporter assay in a condition where SOX9 was expressed under cytomegalovirus (CMV) promoter (i.e., the promoter not affected by SOX9) in HEK293T cells. In this context, tankyrase inhibition using siRNAs or drugs increased the activity of the exogenously expressed SOX9 (Fig. 5f, g). The relevant data are described in the Results (page 9, 2nd paragraph).

[CHX experiment using a weak promoter for SOX9 expression]

Thank you for your valuable suggestion. As the reviewer pointed out, despite the high stability of SOX9 TBD1/2 deletion mutant, we observed that the protein level of SOX9 and SOX9 TBD1/2 deletion mutant showed insignificant difference prior to CHX treatment when they were expressed under the CMV promoter. As suggested by the reviewer, we have subcloned FLAG-tagged SOX9 and SOX9 TBD1/2 deletion mutant into the expression vector with a thymidine kinase (TK) promoter having a relatively weaker promoter activity than that of the CMV promoter (Damdindorj *et al*, *PloS One* 2014, 9(8): e106472). We transfected HEK293 cells with these SOX9 vectors that were further diluted with the control TK vector cloned with *Renilla* luciferase at 1:20 ratio. These procedures ensured much lower and presumably more physiologically relevant expression levels of FLAG-tagged wild type and mutant SOX9 in HEK293 cells. Under this condition, the protein level of SOX9 TBD1/2 deletion mutant was much higher than the protein level of wild-type SOX9 at 0 h, and SOX9 TBD1/2 deletion mutant was degraded more slowly than wild-type SOX9 was (Response Fig. 3b). The relevant data are described in the Results (page 8, 1st paragraph).

[The effect of tankyrase inhibitor on the ubiquitination of SOX9]

The reviewer has raised a valid concern. To address this issue, we examined the effect of tankyrase inhibition on SOX9 ubiquitination. FLAG-tagged human SOX9 was expressed in HEK293 cells treated with XAV939 or vehicle (DMSO). The cells were treated with MG132 6 h before lysis, and the exogenously expressed SOX9 was then pulled down using anti-FLAG M2 magnetic beads. The overexpressed SOX9 underwent extensive poly-ubiquitination, whereas XAV939 treatment nearly abolished the ubiquitination of SOX9 in HEK293 cells (Response Fig. 3c), supporting that tankyrase-mediated PARylation is an essential modification that leads to SOX9 ubiquitination and degradation. The relevant data are described in the Results (page 8, 2nd paragraph).

[E3 ligase mediating the ubiquitination of SOX9]

The reviewer is correct that, to date, RNF146 is the only E3 ubiquitin ligase known to form a complex with tankyrase and recognize PARylated substrates (Andrabi *et al*, *Nat. Med.* 2011, 17(6): 692; Zhang *et al*, *Nat. Cell Biol.* 2011, 13(5): 623; DaRosa *et al*, *Nature*, 2015, 517(7533): 223). However, our results indicated that the effects of tankyrase on SOX9 is not dependent on RNF146 (Fig. 4), suggesting an intriguing possibility that PAR-dependent E3 ligases, other than RNF146, may exist and regulate PARylation-dependent SOX9 activity. From the quantitative mass spectrometry data, we identified additional tankyrase-binding E3 ubiquitin ligases. Our analysis of endogenous tankyrase interactomes revealed that thyroid hormone receptor interactor 12 (TRIP12) and HECT, UBA, and WWE domain-containing 1 (HUWE1) form complexes with tankyrase in chondrocytes. Intriguingly, these E3 ligases contained WWE domains that have the potential to mediate interactions with PARylated substrates (Zhang *et al*, *Nat. Cell Biol.* 2011, 13(5): 623). We are currently pursuing the idea to identify a potential, novel E3 ligase that specifically recognizes PARylated SOX9 as a substrate. Extensive biochemical studies are warranted for characterizing a new class of E3 ubiquitin ligase that mediates poly-ubiquitination of PARylated substrates. The relevant description was provided in the Discussion (page 14, 3rd paragraph).

Major point 4. SOX9 is a master regulator of chondrogenesis. The finding that knockdown of SOX9 abolished the effect of TNKS inhibitor on cartilage matrix gene expression (Fig. 5i)

does not prove the functional interaction between SOX9-TNKS. The only evidence shown in Fig 5h is based on overexpression and it is somewhat weak. A much better experiment would be in vivo knock in.

[Response]

We agreed with the reviewer that the construction of cells with a knock-in mutant of SOX9 that abolishes the binding with tankyrase (i.e., SOX9 R2A) would clearly demonstrate the functional interaction between SOX9 and tankyrase. Thus, we performed CRISPR/Cas9-mediated knock-in experiments.

SOX9 is a master regulator of chondrogenesis, which displays cartilage-specific expression pattern in the mouse skeletal system (Salminen *et al*, *Arthritis Rheum.* 2001, 44(4): 947-955). In particular, the SOX9-mediated transcription of cartilage matrix genes such as *Col2a1* and *Acan* is restricted in chondrocyte-lineage cells (Ohba *et al*, *Cell Rep.* 2015, 12(2): 229-243). Therefore, for CRISPR/Cas9-mediated knock-in of SOX9 mutant, we first attempted to use primary mouse chondrocytes that have the robust *Sox9* promoter activity. We introduced Cas9/sgRNA ribonucleoprotein and single-stranded donor DNA into chondrocytes using a transfection reagent. We designed sgRNA such that Cas9 cleaves between the nucleotide codes for the 257th and 271st amino acids (Response Fig. 4a). Four donor DNAs were designed to create silent mutations, which would serve as the control and four other donor DNAs were designed to cause the amino acid substitutions, R257A and R271A. These eight donor DNAs destroy the KasI recognition site near the nucleotide codes for the 257th amino acid when incorporated into the genome by homology directed repair (HDR). The DNA segment containing the nucleotide codes for the 257th amino acid was amplified and digested with the KasI restriction enzyme. Cleavage at this KasI site was used to distinguish between wild-type and mutant alleles. However, all the amplified DNA segments from chondrocytes were cleaved by KasI, like the DNA segment of wild-type chondrocytes (Response Fig. 4b). We concluded that with this low transfection and knock-in efficiency in the primary cells, there would be an extremely marginal chance to generate HDR-mediated knock-in chondrocyte population.

Alternatively, we searched for cell lines with a measurable degree of *SOX9* promoter activity. We tested two commonly used human chondrosarcoma cell lines, SW1353 and JJ012.

However, these two cell lines expressed undetectable levels of SOX9 and its target genes *COL2A1* and *ACAN*. Rather unexpectedly, we observed a modest level of SOX9 protein in HEK293 cells although this cell line is completely devoid of chondrogenic characteristics. We performed CRISPR/Cas9-mediated knock-in experiments in HEK293 cells. sgRNAs and donor DNAs were designed to create SOX9 R257A or R271A knock-in cell line and delivered to HEK293 cells (Response Fig. 4c). We attempted to use two different transfection methods that utilize a transfection reagent and electroporation, respectively. After transfection, the cells were trypsinized and seeded in 96-well plates at a density of 1 cell/well to grow single cell-derived colonies. We analyzed the transfected HEK293 cells by PCR followed by KasI (for R257A generation) or SallI (for R271A generation) restriction enzyme digestion. More than 1,000 clones were screened, but only one of them was a knock-in cell clone (heterozygous R257A knock-in) (Response Fig. 4d, e). Unfortunately, the positive clone was not viable after several passages.

Last, as an effort to provide evidence for the functional interaction between SOX9 and tankyrase, we examined ubiquitination of “*endogenous SOX9*” in primary chondrocytes with or without tankyrase inhibition (Response Fig. 4f). Using a tandem ubiquitin-binding entity (TUBE) directed against poly-Ubiquitin (poly-Ub) moieties, endogenous proteome with poly-Ub chains were immunoprecipitated. Western blot analysis revealed that endogenous SOX9 undergoes poly-ubiquitination and that it is degraded by the proteasome in chondrocytes as MG132 treatment led to the accumulation of SOX9 and its poly-ubiquitinated forms. XAV939 treatment effectively reduced the poly-ubiquitinated forms of SOX9 comparable to the level observed in the absence of MG132. These results show that under a physiological condition, SOX9 undergoes degradation through the ubiquitin-proteasome system, which can be effectively reversed by the inhibition of tankyrase enzymatic activity. These additional new data are described in the Results of the revised manuscript (page 8, 2nd paragraph).

Reviewer: 2

General comment. Han S et al. reports that tankyrase accelerates OA progression through SOX9 PARylation and the subsequent decrease of cartilage anabolism. They identified tankyrase as a regulator of cartilage anabolic axis by in silico analyses. Expression of Tnks and Tnks2 is inversely correlated with that of Col2a1 and Acan. siRNA against Tnks&Tnks2 or tankyrase inhibitors increase expression of cartilage matrix genes. They further identified SOX9 as a target of tankyrase. Protein-protein binding between SOX9 and tankylase is shown by in situ proximity ligation assay and co-IP. Tankyrase inhibitor enhances SOX9 protein level and expression of SOX9-target genes. Intra-articular injection of tankyrase inhibitor ameliorates OA development in DMM mice. Tankyrase inhibition stimulates chondrogenesis of MSC. The present findings are novel and interesting. Meanwhile, several points should be addressed.

[Response]

Thank you for your careful reading of our manuscript. We appreciate your constructive comments. In the revised manuscript, we have attempted to address your concerns to the best of our ability by performing a series of additional experiments, presenting new experimental data, and modifying the text accordingly.

Major point 1. The authors show that tankyrase inhibition enhanced cartilage anabolism in vitro and in vivo. However, they did not show how tankyrases regulate chondrocytes in OA pathophysiology and chondrogenesis. Is expression of Tnks and Tnks2 increased with OA development? Are they increased or activated in the early stage, or in the late stage? Are they correlated with downregulation of matrix genes in vivo? Are they expressed or activated during chondrogenesis? These analyses are necessary to understand physiological roles of tankyrases in chondrocyte differentiation and articular cartilage homeostasis. These data will be indispensable and much useful for clinical application of tankyrase inhibition in future.

[Response]

We attempted to address these questions by examining the expression of tankyrase in the OA cartilage of human and mouse.

[Tankyrase expression in human OA patients]

Human OA cartilage specimens were sourced from patients with OA undergoing total knee replacement at the Seoul National University (SNU) Boramae Medical Center. The Boramae Medical Center Institutional Review Board (IRB No. 30-2017-48) approved the collection of the human specimens and the SNU Institutional Review Board (IRB No. E1803/003-009) approved the use of these materials. The damage of OA-affected human cartilage was established by Alcian blue staining (Response Fig. 5a). Tankyrase proteins were markedly upregulated in OA-affected cartilage but were barely detectable in undamaged regions of arthritic cartilage ($n = 5$) (Response Fig. 5a, b). In contrast, type II collagen and aggrecan expression was significantly reduced in the damaged region of human OA cartilage, inversely correlating with the expression pattern of tankyrase (Response Fig. 5a).

Presumably due to a high similarity in amino acid sequences between TNKS and TNKS2, we were not able to find tankyrase antibodies that exclusively detect either TNKS or TNKS2. Therefore, we used the TNKS/2 antibody that simultaneously detects both TNKS and TNKS2 (Santa Cruz Biotechnology, sc-365897).

[Tankyrase expression in the early and late stage OA mouse models]

We used destabilization of the medial meniscus (DMM) surgery as a mouse model of post-traumatic OA. Two weeks after DMM, we observed no distinct damage in the cartilage (Kim *et al*, *Proc. Natl. Acad. Sci. U.S.A.* 2015, 112(30): 9424-9429), but detected a robust expression of SOX9 and aggrecan, consistent with the findings of studies that have reported compensatory synthesis of matrix molecules during the “early” stage of OA (Goldring and Goldring, *J. Cell Physiol.* 2007, 213(3): 626-34; Venkatesan *et al*, *PloS One* 2012, **7**(3): e34020) (Response Fig. 5c). At this stage, tankyrase expression was not detected in the knee joint cartilage of the DMM-operated mice. However, 8 weeks after DMM surgery, significant damage in the articular cartilage and loss of SOX9 and aggrecan expression were apparent, indicating the mid-to-late stage of OA (Zhang *et al*, *Osteoarthr. Cartil.* 2015, 23(12): 2259-2268). A marked increase in tankyrase expression was detected in this condition, consistent with our observations in human OA cartilage.

[Tankyrase expression during chondrogenesis]

The expression pattern of *Tnks* and *Tnks2* mRNAs during chondrogenesis was examined using micromass culture of mesenchymal cells isolated from the limb-bud of E11.5 mouse embryos. With the progression of in vitro chondrogenesis, we noted a gradual increase in *Acan* expression, a marker gene for chondrogenesis (Response Fig. 5d). In contrast, *Tnks* and *Tnks2* expression was markedly decreased during chondrogenesis, inversely correlating with the chondrogenic gene expression. We also analyzed the publicly available RNA-Seq dataset generated from pellet cultures of bone marrow-derived MSCs (GSE109503). The expression of chondrogenic markers such as *COL2A1* and *ACAN* was significantly increased during the course of pellet culture; *TNKS* and *TNKS2* expression was markedly reduced (Response Fig. 5e). These results are in line with our experimental results from the micromass culture of mouse limb-bud mesenchymal cells.

These additional new data are described in the Results section of the revised manuscript (page 10, 1st paragraph; page 11, 2nd paragraph).

Major point 2. They use XAV939 and IWR-1 as tankyrase inhibitors. As they describe, they are widely used as inhibitors of WNT/beta-catenin pathway. XAV939 and IWR-1 enhance beta-catenin degradation through stabilization of Axin. Notably, many previous studies have shown that WNT/beta-catenin is a representative pathway for cartilage degradation, and remarkably suppresses chondrogenesis. Sox9 and WNT/beta-catenin inhibit mutual transactivation (Genes Dev 2004 18:1072, and many following papers). These points should be further analyzed in major experiments in the present study. They show that siCtnnb1 and iCRT did not increase matrix genes (Fig2eg), but these data did not exclude that anabolic effects of XAV939 and IWR-1 are independent of inhibition of WNT/beta-catenin. Fig5j did not exclude the possibility that XAV939 enhanced SOX9 via suppression of WNT/beta-catenin. Considering that tankyrase has many kinds of substrates, it is possible that tankyrase inhibitor exerts anabolic effects in chondrocytes through both SOX9 enhancement and WNT inhibition, as the authors described in Discussion. I believe that multifaceted examinations will make the present study more valuable.

[Response]

You have pointed out a valid issue. In fact, reviewer #1 provided similar comments (major point #1) and suggested several experiments to address this concern.

Tankyrase inhibitors conventionally serve as inhibitors of Wnt signaling by preventing Axin degradation (Huang *et al*, *Nature*. 2009, 461(7264): 614; Chen *et al*, *Nat. Chem. Biol.* 2009, 5(2): 100), and Wnt/ β -catenin signaling has been shown to suppress chondrogenic gene expression (Day *et al*, *Dev. Cell* 2005, 8(5): 739-750; Hwang *et al*, *J. Biol. Chem.* 2004, 279(25): 26597-26604). We tested whether chondrogenic genes augmented by tankyrase inhibition is dependent on its inhibitory effect on Wnt/ β -catenin signaling. We explored this issue by using Dkk-1, which antagonizes canonical Wnt ligands and IWP-2, a porcupine inhibitor that blocks the secretion of canonical and noncanonical Wnt.

Chondrocytes treated with tankyrase inhibitors or tankyrase-targeting siRNAs augmented the expression of cartilage matrix genes such as *Col2a1* and *Acan* in the absence of exogenous Wnt treatments. We questioned whether this result is associated with the effect of Wnt/ β -catenin signaling inhibition. However, the treatment of chondrocytes with Dkk-1 or IWP-2 did not affect the expression of *Col2a1* and *Acan* (Response Fig. 1a, b), suggesting that Wnt/ β -catenin signaling has a negligible effect on cartilage anabolism in our experimental context (i.e., in the absence of exogenous Wnt ligands).

We then conducted our experiments in the presence of exogenous Wnt ligands. Treatment of chondrocytes with Wnt-3a, a canonical Wnt ligand, significantly reduced the expression of *Col2a1* and *Acan* (Response Fig. 1a, b). The addition of recombinant Dkk-1 effectively abolished the Wnt-3a-mediated suppression of these chondrogenic genes. Meanwhile, IWP-2 did not rescue the expression of *Col2a1* and *Acan*, consistent with its inability to block exogenously added Wnt ligands. Therefore, in the presence of abundant Wnt ligands, tankyrase inhibition might exert an additional effect to promote cartilage matrix anabolism in a Wnt/ β -catenin-signaling-dependent manner. Nonetheless, our investigation was mainly conducted in a context where β -catenin signaling plays a marginal role in cartilage matrix gene regulation, suggesting that the tankyrase-PARylation-SOX9 axis can regulate the anabolic pathway independent of Wnt/ β -catenin signaling.

Our new data with the Wnt inhibitors are described in the Results section of the revised manuscript (page 6, 3rd and 4th paragraphs).

Major point 3. Although many data are displayed in Figures, some of them are not well written in the text and the legends. I recommend that the authors describe each result more specifically.

[Response]

We appreciate your in-depth examination of our manuscript. Per your suggestion, we have provided a more detailed description of our results in the text and figure legends, especially those of Fig. 1 and 3. Therefore, the word count of the results section and figure legends has increased from 1940 words to 2838 words and from 2035 words to 2672 words, respectively. Please note that the revised text in the Results and figure legends are marked in blue font.

Major point 4. All western blot and IHC should be properly quantified.

[Response]

We quantified WB data based on the band pixel intensity and IHC data based on the total chromogen intensity in a defined area. The quantification procedures are described in the Methods (page 57, 3rd paragraph; Page 63, 2nd paragraph). We have provided the quantification data in the following figures.

WB: Fig. 1f, h, j, Fig. 2b, d, Fig. 3n-p, and Fig. 4b, e, f

IHC: Fig. 6a, b, h, i, Fig. 7i, k, l, Supplementary Fig. 5c, and Supplementary Fig. 6a, b

Major point 5. In Fig6, WNT/beta-catenin-related molecules such as beta-catenin or axin should be examined, as well as Col2, Acan, and Sox9. Tnks and tnks2 should be examined in these series Fig6fg, the authors should display which regions were shown. Are they tibial cartilage or femoral, medial or lateral?

[Response]

[Expression of β -catenin in post-traumatic OA cartilage]

We examined the expression of β -catenin in post-traumatic OA cartilage caused by DMM surgery in mice. Consistent with the findings of various studies from the OA fields (Koyama *et al*, *Dev. Biol.* 2008, 316(1): 62-73; Yasuhara *et al*, *Lab. Invest.* 2011, 91(12): 1739), β -

catenin was specifically expressed in the superficial zone of articular cartilage (Response Fig. 6a). Intra-articular administration of the tankyrase inhibitors, XAV939 or IWR-1, effectively abolished the expression of β -catenin in the superficial zone of post-traumatic OA cartilage.

[The effect of tankyrase inhibitors on tankyrase expression level]

We agree that it is important to assess tankyrase activity in the OA cartilage treated with XAV939 or IWR-1 in comparison with vehicle control. However, it is well established that tankyrase undergoes auto-PARylation, and it is subsequently recognized by RNF146 E3 ligase (Zhang *et al*, *Nat. Cell Biol.* 2011, 13(5): 623; DaRosa *et al*, *Nature*, 2015, 517(7533): 223). Therefore, tankyrase inhibitors cause the accumulation of tankyrase proteins. Consistent with these references, we observed that in primary chondrocyte culture, XAV939 treatment similarly increased the protein level of tankyrase (Response Fig. 6b). Therefore, we did not assess the tankyrase level in OA cartilage treated with tankyrase inhibitors because its level is not a measure of tankyrase activity. Rather, the reduced β -catenin level in OA cartilage treated with XAV939 or IWR-1 indicates that tankyrase activity was effectively abolished in these tissues (Response Fig. 6a).

[Articular cartilage region displayed in the IHC images of Fig. 6]

All the IHC images were acquired from the medial tibial plateau cartilage, which is most susceptible to osteoarthritic damage induced by DMM. The only exception was the IHC against β -catenin, whose expression is restricted to the superficial zone of articular cartilage. As the medial femoral condyle better represents the zonal architecture of mouse articular cartilage including the superficial zone, this region was used for β -catenin imaging. In the Methods, we have stated which region of the knee joint was used for each IHC imaging.

Major point 6. Fig.7a-c, it is not appropriate to determine chondrogenesis only by alcian blue staining, because it is less specific. Expression of marker genes should be also examined. These experiments are suitable to evaluate SOX9 and WNT/beta-catenin pathway.

[Response]

We are grateful to your suggestion. As the reviewer pointed out, Alcian blue is not exclusively specific for sulfated proteoglycans as it has some residual affinity to carboxylated polysaccharide chains even at a very low pH (Gruber *et al*, *Biotech. Histochem.* 2002,

77(2):81-3). Therefore, per the reviewer's suggestion, we examined the expression of chondrogenic marker genes. First, the expression pattern of *Col2a1* and *Acan* mRNAs was examined using a micromass culture of mesenchymal cells in the absence or presence of tankyrase inhibitor, XAV939 or IWR-1 (Response Fig. 7a). Tankyrase inhibitors significantly promoted *Col2a1* and *Acan* expression. In contrast, the PARP1/2 inhibitor, ABT-888, failed to increase their expressions, suggesting that the chondrogenic effect is specific to tankyrase inhibition, but not other PARP family member inhibition. Similarly, in the pellet culture of hMSCs, both pharmacological inhibition and knockdown of *TNKS/2* effectively enhanced chondrogenic markers such as Type II collagen and aggrecan (Response Fig. 7b, c). SOX9 expression was also markedly increased in the presence of tankyrase inhibition. These results and the relevant description have been provided in the revised manuscript (page 11, 3rd paragraph).

Major point 7. Fig.7e-h, *Tnks*, *tnks2*, and WNT/beta-catenin-related molecules should be examined. Additionally, the authors should demonstrate that the regenerative tissues are derived from hMSC, e.g., by IHC using human-specific antibodies.

[Response]

We conducted IHC analysis using a human-specific mitochondrial antibody. No positive signal was detected in osteochondral lesion filled with fibrin gel only (vehicle control), whereas defects transplanted with hMSCs-control shRNA or hMSCs-*TNKS/2* shRNA displayed strong signals localized in the cytoplasm of transplanted cells (Response Fig. 8). We confirmed that the regenerative tissues derived from hMSCs-*TNKS/2* shRNA exhibited the stable knockdown of tankyrase protein. In contrast, the regenerative fibrocartilage originated from hMSCs-control shRNA showed robust expression of tankyrase. Finally, β -catenin expression was very marginal in lesion implanted with hMSCs-control shRNA or hMSCs-*TNKS/2* shRNA. This observation appears to be consistent with the findings of previous studies (Koyama et al, *Dev. Biol.* 2008, 316(1): 62-73; Yasuhara et al, *Lab. Invest.* 2011, 91(12): 1739) and our data that β -catenin is specifically expressed only in the superficial zone articular chondrocytes (Response Fig. 6a). The relevant data are described in the Results (page 11, 4th paragraph).

Minor point 1. In the 1st paragraph of Results, in silico analyses performed in Fig 1ab should be described more precisely and concretely.

[Response]

We have revised the text to more precisely describe the procedures and results of the in silico analyses performed in Fig. 1a, b. The revised text is as follows: “To screen for a key regulatory factor that could be targeted to stimulate cartilage matrix anabolism, we conducted genetic analysis on transcriptomes of mouse reference populations using post-hoc factor analysis. First, we assessed the transcriptional variance in the cartilage tissues of 16 strains of BXD mice²⁵. We noted that, among 21 cartilage matrix genes listed up by Heinegard and Saxne²⁶, 14 cartilage matrix genes showed strong positive correlation in their transcript abundance (Fig. 1a). These high correlations were absent in organs without cartilaginous functions, such as bone femur, kidney, lung, and brain (Supplementary Fig. 1a). A genome-wide co-expression analysis using BXD cartilage transcriptome datasets similarly resulted in the identification of a highly correlated cluster enriched with cartilage matrix genes (Supplementary Fig. 1b). We then attempted to extract a common axis underlying cartilage anabolism by performing a principal component analysis on 14 highly inter-correlated cartilage matrix genes (see black box in Fig. 1a). The first axis identified (Factor 1) essentially reflects the state of cartilage matrix anabolism (Fig. 1b). Next, we used the Gene Ontology (GO) Resource to annotate the 16,074 genes included in the cartilage transcriptome, of which 6,824 belonged to the GO terms: “catalytic activity”, “DNA-binding transcription factor activity”, or “regulation of signaling”, which can potentially have regulatory functions on the collective expression of a set of genes. To focus on the genes with unknown functions in cartilage, we then removed 417 genes that belong to cartilage-related ontologies from our consideration. As a result, a total of 6,407 candidate genes that have a regulatory function, but with unknown roles in cartilage, were selected. We then computed Pearson’s correlation coefficients between these 6,407 genes and Factor 1 genes. Tankyrase showed striking negative correlations with the anabolic axis and with individual cartilage matrix genes and was therefore, investigated further (Fig. 1b, c and Supplementary Fig. 1c, d).” A detailed description of experimental procedures is provided in the Methods of our revised manuscript.

Minor point 2. I recommend that qPCR and western blot are to be displayed in separate subpanels, for example, Fig. 1e,f,g.

[Response]

As per your suggestion, we have presented the qPCR and western blot data in separate subpanels in Fig. 1e, f, g, h, i, j.

Minor point 3. Please check the figure legends of Fig3f and g.

[Response]

Thanks for pointing out the errors. We have switched the legends of Fig. 3f and g so that they now correspond to the relevant figures.

Reviewer: 3

General comment. Using mouse datasets, authors identified tankyrase as a regulator of the cartilage anabolism axis. Tankyrase appears an interesting druggable target since its inhibition leads to SOX9 stabilization with concurrent anabolic effects on articular cartilage. Most likely, SOX9 stabilization is the result of an uncoupling of SOX9 from a poly(ADP-ribosyl)ation (PARylation)-dependent protein degradation pathway. The authors have done a lot of work. The manuscript is well-written and results are very interesting given the fact that, as mentioned, ‘it has been a difficult task [to reconstruct cartilage matrix in osteoarthritic joints] as adult cartilage exhibits marginal repair capacity’.

[Response]

We thank the reviewer for the evaluation of our manuscript. We found all comments to be constructive and helpful as we revised our manuscript.

Major point 1. My main concern relates to the initial analyses and approach taken. Results lines 79-80 ‘We noted that, among 21 cartilage matrix genes listed up...’: why did the authors focus on these 21 genes? That is not completely clear from the Methods section. If a genome wide approach was taken using available microarray gene expression datasets it is a pity to only select these 21 genes to follow-up?

[Response]

We appreciate the reviewer’s comment. The load-bearing function of articular joints is mainly derived from the unique composition of cartilage-specific matrix molecules (Heinegård and Saxne, *Nat. Rev. Rheumatol.* 2011, 7(1): 50). Loss of cartilage-specific collagen components such as *Col2a1* (Hytinen *et al*, *Ann. Rheum. Dis.* 2001, 60(3): 262-268; Lapveteläinen *et al*, *Osteoarthr. Cartil.* 2001, 9(2): 152-160) and *Col9a1* (Stolz *et al*, *Nat. Nanotechnol.* 2009, 4(3), 186) or proteoglycan components such as *Acan* (Zhang *et al*, *Osteoarthr. Cartil.* 2015, 23(12): 2259-2268) and *Matn3* (van der Weyden *et al*, *Am. J. Pathol.* 2006, 169(2): 515-527) play a key role in OA progression. The 21 genes that we focused on are actually the well-established list of cartilage matrix genes that are essential for the mechanical function of articular cartilage (Heinegård and Saxne, *Nat. Rev. Rheumatol.* 2011, 7(1): 50).

Nonetheless, we agreed with the reviewer that it is worth conducting a *genome-wide* co-expression analysis using BXD cartilage transcriptome datasets having expression profiles of 16074 genes. We were able to identify a highly correlated module enriched with cartilage matrix genes (Response Fig. 9a). Among the 14 cartilage matrix genes that were regulated by tankyrase inhibition, 11 of them were contained in this geneset (hypergeometric $P < 1.53 \times 10^{-7}$). Next, we performed a principal component analysis using the transcriptome profiles of all genes in this module. The first principal component (PC1) essentially reflects the collective expression levels of the module genes. We computed Pearson's correlation coefficients between PC1 and tankyrase. Both *Tnks* and *Tnks2* showed negative correlation with the PC1 of the gene module (Response Fig. 9b), in a similar correlation pattern observed in Fig. 1c.

Our results from the genome-wide co-expression analysis using the BXD cartilage transcriptome are described in page 5, 1st paragraph of the revised manuscript.

Major point 2. Results line 85 'We computed Pearson's correlation coefficients between Factor 1 and various genes...': how many and which genes are these? How were the genes selected?

[Response]

We used the Gene Ontology (GO) Resource to annotate the 16074 genes of the cartilage dataset, of which 6824 genes belonged to the GO terms: "catalytic activity", "DNA-binding transcription factor activity", or "regulation of signaling", which can potentially have a regulatory function for the collective expression of a set of genes. We then removed 417 genes that belong to cartilage-related ontologies from our consideration. As a result, a total of 6407 candidate genes that have a regulatory function with unknown roles in cartilage were selected. We then computed Pearson's correlation coefficients between Factor 1 and the 6407 genes (Response Fig. 10). Among the 6407 genes, *Tnks2* had the 426th highest Pearson's correlation coefficient ($r = 0.841$). We have described this more clearly in the revised manuscript (page 5, 1st paragraph).

Major point 3. Methods lines 521-523 'For the cartilage and bone femur data sets, a probe having non-overlapping SNPs, 'Perfect' quality, and the highest expression was used for each

transcript’: what do the authors mean with ‘non-overlapping SNPs’? And what is the definition of ‘perfect quality’?

[Response]

We used *illuminaMousev1.db* and *illuminaMousev1p1.db* to annotate probes in datasets generated from mouse cartilage and bone femur. First, the two databases provide information about the quality grade for each probe with the following criteria.

“Perfect” if it perfectly and uniquely matches the target transcript; “Good” if the probe, although imperfectly matching the target transcript, is still likely to provide a considerably sensitive signal (up to two mismatches are allowed, based on empirical evidence that the signal intensity for 50-mer probes with less than 95% identity to the respective targets is less than 50% of the signal associated with perfect matches *); “Bad” if the probe matches repeat sequences, intergenic or intronic regions, or is unlikely to provide a specific signal for any transcript; “No match” if it does not match any genomic region or transcript (Dunning *et al*, R package version 1.26.0. 2015).

Next, we wanted to rule out any possibility that SNPs within each probe sequence contributed to the differential expression among 16 BXD mouse strains as a result of altered hybridization efficiency. Therefore, we only used the probes that did not overlap with any known SNPs, which were defined as ‘non-overlapping SNPs’.

We have now clearly described the above information regarding ‘perfect’ quality of probes and ‘non-overlapping SNPs’ in the Methods section (page 56, 1st paragraph).

Minor point 1. Results line 87 ‘...negative correlations with this anabolic axis’ and line 203 ‘... indicating that tankyrase exhibits a positive correlation with ... catabolic regulators’: what about *Mmp13*?

[Response]

In the Supplementary Fig. 4c of the original manuscript, we provided a heatmap of Pearson’s correlation coefficients between transcript levels of *Tnks* or *Tnks2* and *Mmp13* in the articular cartilage of 16 BXD mouse strains. Although *Mmp13* had overall positive correlation with

Tnks and *Tnks2* as indicated by the red colors in the heatmap, there was no statistical significance (i.e., for both *Tnks* and *Tnks2*, $P > 0.05$).

Minor point 2. Results line 100 ‘were similarly clustered’ please explain, what do you mean?

[Response]

Thanks for requesting further clarification regarding our statement: “*Differentially expressed genes (DEGs) in the three tankyrase inhibition groups (compared to respective control groups) were similarly clustered*”. We agree that we should have provided a more quantitative description regarding this observation. We have now provided the Venn diagram for the DEGs of the three different tankyrase inhibition scenarios (i.e., si*Tnks* + si*Tnks2* vs. siControl, XAV939 vs. DMSO, and IWR-1 vs. DMSO). In fact, most DEGs were commonly upregulated or downregulated by both genetic suppression and pharmacological inhibition of tankyrase. Of the 711 DEGs, 74.12% (527 genes) were consistently increased or decreased in all three tankyrase inhibition groups (Response Fig. 11). We have now provided the relevant description on page 6, 2nd paragraph of the revised manuscript.

Minor point 3. Results line162 ‘We explored whether tankyrase inhibition affects the transcriptional activity of SOX9...’. Maybe I missed something, but previously authors showed ‘the disruption of the physical interactions between tankyrase and SOX9 ... results in the stabilization of SOX9’ (lines 146-147). Isn’t it logical that increased SOX9 levels result in the observed increased transcriptional activity of SOX9? Can the authors show a Western blot for the expression levels of SOX9?

[Response]

The reviewer is correct in that the increased stability of SOX9 by tankyrase inhibition results in the increase in SOX9 levels, which we demonstrated in Fig. 3n, o by conducting western blot analyses. Similarly, the mutations in the tankyrase-binding sites in SOX9 enhanced SOX9 stability and protein levels as shown in western blot analyses (Fig. 3p). As the reviewer mentioned in his/her comment, upregulated SOX9 appears to be the cause for the increased transcriptional activity of SOX9 in the reporter gene assay. A major reason we conducted this assay was to demonstrate that stabilized SOX9 by tankyrase inhibition is actually the functional form of SOX9, which can be engaged in enhancing the transcriptional

activation of target genes. Since our statement: “We explored whether tankyrase inhibition affects the transcriptional activity of SOX9...” may have been confusing, we modified the text accordingly. The revised text now reads: “We used a 4×48-p89 SOX9-dependent *Col2a1* enhancer reporter to investigate whether the increase in SOX9 levels with tankyrase inhibition enhances the overall transcriptional activity of SOX9.”

Minor point 4. Results line 171 ‘...regulation of SOX9 occurs at the protein level rather than at transcription level...’.

[Response]

Although we demonstrated that tankyrase inhibition causes SOX9 stabilization and its accumulation at the protein level, increased SOX9 can in turn promote *SOX9* mRNA transcription by binding to its own enhancer (Mead *et al*, *Nucleic Acids Res.* 2013, 41(8): 4459-4469). Therefore, we attempted to decouple the regulation of SOX9 at the protein level from this auto-regulatory mechanism at the transcriptional level to account for the increased SOX9 levels following tankyrase inhibition. Since the current description might have been confusing, we modified the text to deliver our meaning more clearly. The modified text now reads: “To further confirm the proposed mechanism that tankyrase regulates SOX9 activity post-transcriptionally, we repeated the luciferase reporter assays using the SOX9-dependent *Col2a1* enhancer construct in cells constitutively expressing *SOX9* mRNA under the control of a cytomegalovirus (CMV) promoter.”

Minor point 5. It is known that loss of tankyrases result in high microsatellite instability due to deregulated poly(ADP-ribosyl)ation-mediated signaling which might contribute to development of cancers with high microsatellite instability (PMID:21333322). Are the authors not concerned this will affect joint tissues when applying tankyrase inhibitors as a treatment for OA?

[Response]

Thank you for the comment. As the reviewer mentioned, safety (not just efficacy) is a critical consideration for the development of disease-modifying OA drugs. Any potential toxicity of tankyrase inhibitors should be carefully assessed during preclinical and clinical phases. Nevertheless, the role of tankyrase loss in causing microsatellite instability appears to require

further validation. Kim et al. (*Hum. Pathol.* 2011, 42(9):1289-1296; PMID:21333322) showed the correlation between the occurrence of *TNKS* or *TNKS2* mutations and high microsatellite instability in cancers. However, their ‘functional’ linkages were not explored in this study. Moreover, because *TNKS* and *TNKS2* are known to be functionally redundant (Huang et al, *Nature* 2009, 461(7264): 614; Chiang et al, *PloS One* 2008, 3(7): e2639), a frameshift mutation in either of *TNKS* or *TNKS2* is actually unlikely to abolish overall tankyrase activity. Indeed, in our study, knockdown of either *Tnks* or *Tnks2* activity was insufficient to elicit downstream effects of tankyrase inhibition (Fig. 1e). Taken together, further investigation is needed to evaluate the potential safety issues of tankyrase inhibitors, including its potential association with microsatellite instability.

Minor point 6. At some places the logical order of the text could be improved. For example:

- a) In line 114 the tankyrase-RNF146 complex is introduced and authors mention ‘Consistently, tankyrase inhibition reduced β -catenin stability...’ and not until line 150 the authors explain ‘RNF146 is best known to regulate tankyrase-dependent Axin degradation and, hence, β -catenin stabilization’.
- b) In line 161 it is mentioned that ‘SOX9, a master transcription factor of chondrogenesis, transcribes various cartilage-specific matrix genes’. But SOX9 is introduced already in line 110.

[Response]

We appreciated the reviewer’s comment and agreed that the logical order of some of the text should be reconsidered. In the revised manuscript, we have extensively reorganized our figures and text, focusing on a more logical sequence. Please also note that the suggestions you have made regarding a) and b) have also been taken into account. All of our changes have been highlighted in blue in the revised manuscript.

Reviewers' Comments:

Reviewer #1:

Remarks to the Author:

The quality of the revised manuscript is significantly improved.

Major point: TNKS dependent degradation of SOX9 is a major point of the manuscript.

Demonstration of this point is complicated by the fact that SOX9 increases its own transcription. In the revised manuscript, authors have shown that Flag-SOX9 under control of a weak promoter has lower stability than SOX9 TBD deletion mutant (Fig. 3p). A straightforward follow-up experiment is to test whether TNKS inhibitor can increase the protein level of wild-type SOX9, but not that of TBD deletion mutant, in the same experimental setting (authors performed transient transfection, but a lentivirus-based system might minimize variation of transfection efficiency). I feel that this rather simple experiment will significantly improve readers' confidence of this major conclusion of the manuscript.

Reviewer #2:

Remarks to the Author:

The manuscript has been well refined. The authors provided "Response Fig" in the point-by-point response, but it seems confusing. It would be better to describe simply where the data are shown in the revised manuscript in the point-by-point responses.

1. I cannot find Response Fig 5c in the revised manuscript. I recommend they should be displayed in Figures or Supplementary Figures.
2. In the response to Major point 2, the direct data indicating the status or effect of canonical Wnt signaling under tankyrase inhibitor treatment should be added. Treatment with tankyrase inhibitors in siCtnnb1-transfected chondrocytes would strengthen the present hypothesis. WB of intra-nuclear beta-catenin in chondrocytes treated with tankyrase inhibitors would be also helpful.
3. Fig. 7i, IHC of human mitochondria, images including negative areas (i.e., host tissue) are necessary. Lower magnification images should be additionally shown. Without these data, they cannot determine whether IHC is trustworthy or not.

Reviewer #3:

Remarks to the Author:

I thank the authors for their thorough work and revision in response to the reviewers' comments. To my knowledge, all questions have been answered well and results support the conclusions.

I have only two minor remarks/suggestions:

1. Although results support the benefit of Tankyrase inhibition for inhibition of cartilage destruction, it should be noted that Tnks2 is ranked nr. 426 in the correlation, and Tnks is even less well correlated. To support their selection of this target, authors may want to mention whether or not this is the highest ranked gene in the Wnt-pathway (known to play an important role in cartilage homeostasis). Additionally, the authors can make reference to the paper of Bastakoty et al (doi: 10.1096/fj.15-275941) who previously showed the benefit of XAV-939 in regenerative repair of cartilage.
2. With respect to Supplementary Figure 1d. Authors show the top 10 ranking for Pearson's correlation coefficients between Factor 1 and various genes. In the legend they state that these have 'a regulatory function with unknown roles in cartilage'. I suggest to delete the latter: PLOD3 (ranked nr. 3) encodes a lysyl hydroxylase enzyme (LH/PLOD) and is essential for collagen

biosynthesis. As such, it may also play a role in cartilage extracellular matrix composition.

Point-by-point response to the reviewer's comments

Reviewer: 1

The quality of the revised manuscript is significantly improved.

Major point: TNKS dependent degradation of SOX9 is a major point of the manuscript. Demonstration of this point is complicated by the fact that SOX9 increases its own transcription. In the revised manuscript, authors have shown that Flag-SOX9 under control of a weak promoter has lower stability than SOX9 TBD deletion mutant (Fig. 3p). A straightforward follow-up experiment is to test whether TNKS inhibitor can increase the protein level of wild-type SOX9, but not that of TBD deletion mutant, in the same experimental setting (authors performed transient transfection, but a lentivirus-based system might minimize variation of transfection efficiency). I feel that this rather simple experiment will significantly improve readers' confidence of this major conclusion of the manuscript.

[Response]

We appreciate again your constructive input towards improving our manuscript. As per the reviewer's suggestion, we generated HEK293 cell lines that stably express human SOX9 wild-type or TBD1/2 deletion mutant via lentiviral transduction. Briefly, we subcloned flag-tagged human SOX9 wild-type (WT) and TBD1/2 deletion (Δ TBD1/2) mutant with TK promoter into the lentiviral vector, pLVX-Puro. HEK293 cells were then transduced with Lenti-TK-FLAG-SOX9-WT or Lenti-TK-FLAG-SOX9- Δ TBD1/2. We selected HEK293-FLAG-SOX9-WT and HEK293-FLAG-SOX9- Δ TBD1/2 by puromycin treatment.

The selected HEK293 stable cells were treated with DMSO or XAV939 (10 μ M) for 72 h and their lysates were analyzed by immunoblotting with anti-FLAG antibody to assess the protein level of exogenously expressed SOX9. XAV939 treatment increased the wild-type SOX9 tagged with FLAG. In contrast, the protein level of SOX9 TBD1/2 deletion mutant was not affected by tankyrase inhibition. The relevant data are presented in Fig. 3p.

Reviewer: 2

The manuscript has been well refined. The authors provided “Response Fig” in the point-by-point response, but it seems confusing. It would be better to describe simply where the data are shown in the revised manuscript in the point-by-point responses.

[Response]

Thank you for your thorough review of our revised manuscript. In this version of our point-by-point response, we explicitly describe where new data are presented in the revised manuscript, instead of providing a separate response figure.

1. I cannot find Response Fig 5c in the revised manuscript. I recommend they should be displayed in Figures or Supplementary Figures.

[Response]

During the revision, we repeated the corresponding experiments 4 times and presented the data in Supplementary Fig. 5. We have now included the quantitation result showing ‘cells positive for tankyrase expression’ with statistical analysis in Supplementary Fig. 5b.

2. In the response to Major point 2, the direct data indicating the status or effect of canonical Wnt signaling under tankyrase inhibitor treatment should be added. Treatment with tankyrase inhibitors in siCtnnb1-transfected chondrocytes would strengthen the present hypothesis. WB of intra-nuclear beta-catenin in chondrocytes treated with tankyrase inhibitors would be also helpful.

[Response]

[The effect of tankyrase inhibition in siCtnnb1-transfected chondrocytes]

In accordance with the reviewer’s suggestion, the expression of cartilage-specific matrix genes in mouse chondrocytes transfected with control siRNA or siCtnnb1 was examined after treatment with DMSO or XAV939 for 72 h. Both *Col2a1* and *Acan* were induced by tankyrase inhibition in chondrocytes transfected with control siRNA. Similarly, these genes were reliably induced by XAV939 treatment in chondrocytes transfected with siCtnnb1, suggesting that β -catenin signaling has a negligible effect on the cartilage anabolism induced by tankyrase

inhibition in our experimental context (i.e., in the absence of exogenous Wnt ligands). The relevant results have been described in the Result section (page 6) and presented in Fig. 2i.

[Nuclear and cytoplasmic fractions of β -catenin in chondrocytes]

In accordance with the reviewer's suggestion, we conducted cytoplasmic and nuclear extraction from mouse chondrocytes treated with DMSO or XAV939. Isolation of cytoplasmic and nuclear fractions was confirmed through immunoblot detection of α Tubulin and histone H3 marker proteins. β -catenin was detected in both cytoplasmic and nuclear fractions even in the absence of exogenous Wnt ligands. XAV939 treatment caused a minor reduction in the protein level of β -catenin in both fractions (Fig. 2a). Its effect on β -catenin stability was less pronounced compared to the effect of tankyrase inhibition observed in the presence of exogenous Wnt ligands (Fig. 2b-e). This observation was consistent throughout the repeat trials (shown below as a Supporting Figure).

The relevant experimental methods and results have been included in the Methods section (page 19) and Results section (page 6) of the revised manuscript. The modified text in the Result section reads: "Tankyrase inhibition caused a minor reduction in the β -catenin protein level in chondrocytes (Fig. 2a); the effects of tankyrase inhibition on β -catenin stability and activity were more pronounced in chondrocytes treated with exogenous Wnt ligands (Fig. 2b-e)"

Supporting Figure: Cytoplasmic and nuclear fractions from chondrocytes treated with DMSO or XAV939 (10 μ M, 72 h) were immunoblotted for β -catenin ($n = 3$). BR indicates biological replicates.

3. Fig. 7i, IHC of human mitochondria, images including negative areas (i.e., host tissue) are necessary. Lower magnification images should be additionally shown. Without these data, they cannot determine whether IHC is trustworthy or not.

[Response]

During the revision, we acquired the IHC images of human mitochondria using the lowest magnification objective (4x) the Nikon Eclipse Ni-U upright microscope is equipped with. No positive “*cellular*” signal was detected in the osteochondral lesion filled with fibrin gel alone (vehicle control) despite the presence of background-level, non-specific signals. In contrast, the regions engrafted with hMSCs-control shRNA or hMSCs-*TNKS/2* shRNA displayed strong “*cellular*” signals localized in the cytoplasm of transplanted cells. The relevant data have been presented in Supplementary Fig. 9a.

Reviewer: 3

I thank the authors for their thorough work and revision in response to the reviewers' comments. To my knowledge, all questions have been answered well and results support the conclusions.

I have only two minor remarks/suggestions:

1. Although results support the benefit of Tankyrase inhibition for inhibition of cartilage destruction, it should be noted that *Tnks2* is ranked nr. 426 in the correlation, and *Tnks* is even less well correlated. To support their selection of this target, authors may want to mention whether or not this is the highest ranked gene in the Wnt-pathway (known to play an important role in cartilage homeostasis). Additionally, the authors can make reference to the paper of Bastakoty et al (doi:10.1096/fj.15-275941) who previously showed the benefit of XAV-939 in regenerative repair of cartilage.

[Response]

Thank you for your comment. Among the Wnt-related genes (431 *Mus musculus* genes in the Gene Ontology class "Wnt signaling pathway"), *Tnks2* was the 12th highest with regard to Pearson's correlation coefficient. We have now presented a table listing the top 20 ranking genes based on Pearson's correlation coefficients between Factor 1 and the Wnt-related genes in Supplementary Fig. 3. The result is described on page 6 of the revised manuscript.

As suggested by the reviewer, we have now cited the study by Bastakoty et al in our manuscript. The modified text now reads: "Various β -catenin inhibitory therapeutics have been used at the pre-clinical (Bastakoty *et al*, *FASEB J.* 2015, 29(12): 4881-4892.; Deshmukh *et al*, *Osteoarthr. Cartil.* 2018, 26(1): 18-27; Lietman *et al*, *JCI Insight* 2018, 3(3)) and clinical stage (Phase 1: NCT02095548; Phase 2: NCT02536833, NCT03122860; Phase 3: NCT03928184), with promising outcomes in delaying cartilage destruction."

2. With respect to Supplementary Figure 1d. Authors show the top 10 ranking for Pearson's correlation coefficients between Factor 1 and various genes. In the legend they state that these have 'a regulatory function with unknown roles in cartilage'. I suggest to delete the latter: PLOD3 (ranked nr. 3) encodes a lysyl hydroxylase enzyme (LH/PLOD) and is essential for

collagen biosynthesis. As such, it may also play a role in cartilage extracellular matrix composition.

[Response]

Thank you for pointing this out. As per your suggestion, we have now changed the legend of Supplementary Figure 1d to: “Pearson’s correlation coefficients between Factor 1 and various genes with a potential regulatory function.” We have also modified our text in the Results section (page 5) accordingly: “Next, we used the Gene Ontology (GO) Resource to annotate the 16,074 genes included in the cartilage transcriptome, of which 6,824 belonged to the GO terms: “catalytic activity,” “DNA-binding transcription factor activity,” or “regulation of signaling,” which can potentially have regulatory functions on the collective expression of a set of genes. We then computed Pearson’s correlation coefficients between Factor 1 and these 6,824 genes with potential regulatory functions.”

Reviewers' Comments:

Reviewer #1:

Remarks to the Author:

Authors have addressed my concerns. I support its publication.

Reviewer #2:

None